# AUC Maximization under Positive Distribution Shift

**Atsutoshi Kumagai**
NTT
atsutoshi.kumagai@ntt.com

**Tomoharu Iwata**
NTT
tomoharu.iwata@ntt.com

**Hiroshi Takahashi**
NTT
hiroshibm.takahashi@ntt.com

**Taishi Nishiyama**
NTT Security Holdings, NTT
taishi.nishiyama@security.ntt

**Yasuhiro Fujiwara**
NTT
yasuhiro.fujiwara@ntt.com

## Abstract

Maximizing the area under the receiver operating characteristic curve (AUC) is a common approach to imbalanced binary classification problems. Existing AUC maximization methods usually assume that training and test distributions are identical. However, this assumption is often violated in practice due to *a positive distribution shift*, where the negative-conditional density does not change but the positive-conditional density can vary. This shift often occurs in imbalanced classification since positive data are often more diverse or time-varying than negative data. To deal with this shift, we theoretically show that the AUC on the test distribution can be expressed by using the positive and marginal training densities and the marginal test density. Based on this result, we can maximize the AUC on the test distribution by using positive and unlabeled data in the training distribution and unlabeled data in the test distribution. The proposed method requires only positive labels in the training distribution as supervision. Moreover, the derived AUC has a simple form and thus is easy to implement. The effectiveness of the proposed method is experimentally shown with six real-world datasets.

## 1 Introduction

In many real-world binary classification problems such as intrusion detection [37], medical diagnosis [60], and visual inspection [42], *class-imbalance* frequently occurs where positive data is much smaller than negative data [22]. In this case, classification accuracy, which is the standard performance measure for ordinary binary classification, is not a suitable measure [54, 61]. Instead, the area under the receiver operating characteristic curve (AUC) is commonly used [4, 19]. The AUC is the probability that a classifier will rank a randomly drawn positive instance higher than a randomly drawn negative one [61]. Due to the nature of the ranking, the AUC can adequately measure the performance of the classifier even with imbalanced data. By maximizing the AUC, we can obtain accurate classifiers even from imbalanced data [5, 61, 64, 32, 65].

Existing AUC maximization methods usually assume that training and test distributions are identical to ensure the generalization performance. However, in real-world applications, this assumption is often violated due to distribution shifts. This paper considers *a positive distribution shift* [16], i.e., the negative-conditional density does not change but the positive-conditional density can vary, because this shift often occurs in imbalanced problems. For example, in intrusion detection, malicious

adversaries rapidly change their attacks (positive data) to bypass detection systems while the benign class's data (negative data) do not change much [9, 16, 67]. In medical diagnosis, the distribution of disease data (positive data) can change due to the disease progression or emergence of new pathogens, but the distribution of data of healthy people is generally stable. In visual inspection, the types of anomalous products (positive data) are diverse but normal ones (negative data) have some degree of stationarity. When such a distribution shift occurs, the performance of AUC maximization methods drastically deteriorates.

Although labeled data drawn from the test distribution can alleviate this problem, such data are often time-consuming and expensive to collect whenever the distribution shift occurs [41]. In addition, labeled negative data in the training distribution are also difficult to collect in some applications. For example, in intrusion detection, although some malicious data can be collected from public sources such as blacklists, benign data are often unavailable due to privacy reasons, and identifying clean benign data from given unlabeled data requires a high level of expertise [37, 47].

In this paper, we propose a method for maximizing the AUC under the positive distribution shift by using labeled positive and unlabeled data in the training distribution and unlabeled data in the test distribution. Figure 1 illustrates examples of given data in our problem setting. Since no labeled data are available in the test distribution, a challenge is how to maximize the AUC on the test distribution. To address this challenge, we theoretically show that the AUC on the test distribution can be expressed by using the positive and marginal training densities and the marginal test density when assuming the positive distribution shift. This result enables us to maximize the AUC on the test distribution with positive and unlabeled data in the training distribution and unlabeled data in the test distribution. The derived AUC has a simple form and is easy to implement. In addition, it does not depend on the class-prior in the test distribution, which is usually difficult to obtain with unlabeled data in

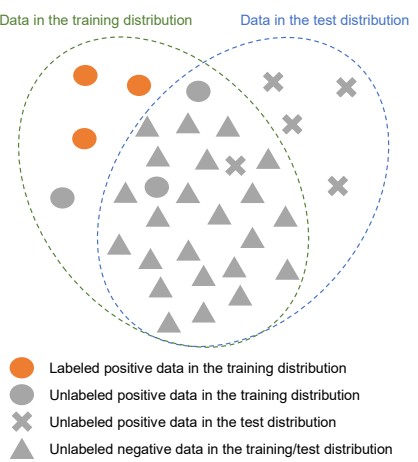

Figure 1: Illustration of given data in our problem setting. The triangle, circle, and cross represents negative data, positive data in the training distribution, and positive data in the test distribution, respectively. Orange and gray represents labeled and unlabeled data, respectively. Our setting assumes that the distribution of negative data does not change (triangles) but that of positive data can vary (circles and crosses).

the test distribution. Also, it can be applied to any differentiable classifiers such as linear classifiers, kernel-based classifiers, and neural networks. Hence, our method is easy for practitioners to use.

The main contributions of our work are summarized as follows:

- We propose a novel and practical problem setting, where the aim is to maximize the AUC under the positive distribution shift on imbalanced data.

- We theoretically show that the AUC on the test distribution can be maximized with positive and unlabeled data in the training distribution and unlabeled data in the test distribution when assuming the positive distribution shift.

- We empirically demonstrate that the proposed method outperformed various existing methods with six real-world datasets.

## 2 Related Work

Many AUC maximization methods have been proposed [5, 61, 64, 32, 65]. In previous studies [65, 66, 12, 55], they have been reported to often perform better than other methods for imbalanced classification such as class balanced loss [6], focal loss [31], or sampling-based methods [8, 36]. However, these AUC maximization methods require labeled positive and negative data. In addition, they assume that the training and test distributions are the same. Therefore, they are inappropriate for our problem setting where there are no labeled negative data and the distribution shift occurs.

Unsupervised domain adaptation aims to adapt to the distribution (domain) shift by using unlabeled data in the test distribution and labeled data in the training distribution [41, 56]. One representative approach is to learn domain-invariant feature representations by minimizing the discrepancy of the features in both domains [33, 53, 52, 13, 27, 11, 48]. Although this approach is promising, it often deteriorates the performance since it minimizes only the feature discrepancy without considering the relationship between features and labels [70]. Another representative approach is to explicitly minimize the loss on the test distribution by assuming the types of the distribution change, such as covariate shift [50, 3, 23, 69, 35] and class-prior shift [68, 51, 1]. The proposed method belongs to this approach. By assuming the shift type, this approach can adapt to the shift in a theoretically guaranteed manner. The existing methods aim to minimize the classification risk (or negative classification accuracy), which is an inappropriate metric for imbalanced data (e.g., if the imbalanced data ratio is $1 : 99$, a naive classifier that classifies 'every' instance as negative has $99\%$ accuracy, but it is definitely not a good classifier). One domain adaptation method tries to maximize the AUC by learning domain-invariant feature representations [62]. However, this method is not designed for the positive distribution shift. Moreover, all these methods require labeled positive and negative data in the training distribution, which are unavailable in our problem setting.

Positive-unlabeled (PU) learning methods aim to learn classifiers by using only positive and unlabeled data [2, 10]. The proposed method is related to the PU learning since it assumes positive and unlabeled data in the training distribution. A representative PU learning method is the empirical minimization-based approach, which rewrites the classification risk by using positive and unlabeled densities [9, 25, 49, 21]. Although they are effective, they cannot maximize the AUC. Recent studies have shown that the AUC can be rewritten from positive and marginal densities by using the technique of the PU learning [45, 58, 6, 59]. However, these all methods assume that the training and test distributions are identical.

Several PU learning methods consider the distribution shift such as covariate shift [46], class-prior shift [7, 39], or positive distribution shift [16]. They require positive and unlabeled data in the training distribution and unlabeled data in the test distribution. However, they consider the classification risk and cannot maximize the AUC. Due to the pairwise formulation of the AUC, their methods that use ordinary instance-wise loss functions such as the cross-entropy loss cannot be applied to the AUC. In addition, the method for the positive distribution shift [16] requires the class-prior on the test distribution, which is generally difficult to know with unlabeled data in the test distribution. In contrast, the proposed method does not need to know the class-prior and thus is more practical.

## 3   Preliminary

We briefly explain the AUC maximization. Let instance $\mathbf{x} \in \mathbb{R}^D$ and its corresponding label $y \in \{-1, +1\}$ be equipped with probability density $p(\mathbf{x}, y)$, where $+1$ and $-1$ means a positive and negative class, respectively. Here, $p^{\mathrm{p}}(\mathbf{x}) := p(\mathbf{x}|y = +1)$ and $p^{\mathrm{n}}(\mathbf{x}) := p(\mathbf{x}|y = -1)$ is the conditional probability density of positive and negative class, respectively. Furthermore, let $s : \mathbb{R}^D \to \mathbb{R}$ be a score function that outputs the positivity of an input instance. The classifier is defined by the score function with threshold $t$: $y = \mathrm{sign}(s(\mathbf{x}) - t)$, where $\mathrm{sign}$ is a sign function.

The AUC is the probability of a randomly drawn positive instance being ranked before a randomly drawn negative instance [61]. Specifically, the AUC with score function $s$ can be formulated as

$$\mathrm{AUC}(s) = \mathbb{E}_{\mathbf{x}^{\mathrm{p}} \sim p^{\mathrm{p}}(\mathbf{x})} \mathbb{E}_{\mathbf{x}^{\mathrm{n}} \sim p^{\mathrm{n}}(\mathbf{x})} \left[ I(s(\mathbf{x}^{\mathrm{p}}) > s(\mathbf{x}^{\mathrm{n}})) \right], \tag{1}$$

$I(z)$ is the indicator function that outputs 1 if $z$ is true and 0 otherwise, and $\mathbb{E}$ is the expectation. Since the gradient of indicator function $I$ is zero everywhere except for the origin, the AUC cannot be optimized via gradient descent methods. To avoid this, the following smoothed AUC is often used by replacing the indicator function with a sigmoid function $\sigma(z) = 1/(1 + \exp(-z))$ [20, 28, 29]:

$$\mathrm{AUC}_\sigma(s) = \mathbb{E}_{\mathbf{x}^{\mathrm{p}} \sim p^{\mathrm{p}}(\mathbf{x})} \mathbb{E}_{\mathbf{x}^{\mathrm{n}} \sim p^{\mathrm{n}}(\mathbf{x})} \left[ \sigma(s(\mathbf{x}^{\mathrm{p}}) - s(\mathbf{x}^{\mathrm{n}})) \right]. \tag{2}$$

Given $N^{\mathrm{p}}$ positive instances $\{\mathbf{x}_1^{\mathrm{p}}, \ldots, \mathbf{x}_{N^{\mathrm{p}}}^{\mathrm{p}}\}$ drawn from $p^{\mathrm{p}}(\mathbf{x})$ and $N^{\mathrm{n}}$ negative instances $\{\mathbf{x}_1^{\mathrm{n}}, \ldots, \mathbf{x}_{N^{\mathrm{n}}}^{\mathrm{n}}\}$ drawn from $p^{\mathrm{n}}(\mathbf{x})$, the empirical estimate of the smoothed AUC is calculated as

$$\widehat{\mathrm{AUC}}_\sigma(s) = \frac{1}{N^{\mathrm{p}} N^{\mathrm{n}}} \sum_{n=1}^{N^{\mathrm{p}}} \sum_{m=1}^{N^{\mathrm{n}}} \left[ \sigma(s(\mathbf{x}_n^{\mathrm{p}}) - s(\mathbf{x}_m^{\mathrm{n}})) \right]. \tag{3}$$

By maximizing this empirical smoothed AUC with respect to the parameters of $s$, we can obtain good score functions to maximize the AUC when the training and test distributions are identical [61].

## 4 Proposed Method

In this section, we first describe our problem setting (subsection 4.1). Then, we theoretically show that the AUC on the test distribution can be maximized with positive and unlabeled data in the training distribution and unlabeled data in the test distribution under the positive distribution shift (subsection 4.2). Then, we discuss class-priors used in the proposed method (subsection 4.3). Lastly, we explain an extension of the proposed method (subsection 4.4).

### 4.1 Problem Setting

Suppose that we are given a set of positive instances $X_{\mathrm{tr}}^{\mathrm{P}}$ and a set of unlabeled instances $X_{\mathrm{tr}}$ drawn from the training distribution:

$$X_{\mathrm{tr}}^{\mathrm{P}} = \{\mathbf{x}_{\mathrm{tr},n}^{\mathrm{P}}\}_{n=1}^{N_{\mathrm{tr}}^{\mathrm{P}}} \sim p_{\mathrm{tr}}^{\mathrm{P}}(\mathbf{x}) := p_{\mathrm{tr}}(\mathbf{x}|y = +1), \tag{4}$$

$$X_{\mathrm{tr}} = \{\mathbf{x}_{\mathrm{tr},n}\}_{n=1}^{N_{\mathrm{tr}}} \sim p_{\mathrm{tr}}(\mathbf{x}) = \pi_{\mathrm{tr}} p_{\mathrm{tr}}^{\mathrm{P}}(\mathbf{x}) + (1 - \pi_{\mathrm{tr}}) p_{\mathrm{tr}}^{\mathrm{n}}(\mathbf{x}), \tag{5}$$

where $p_{\mathrm{tr}}(\mathbf{x})$ is the marginal density of the training distribution, $p_{\mathrm{tr}}^{\mathrm{P}}(\mathbf{x})$ and $p_{\mathrm{tr}}^{\mathrm{n}}(\mathbf{x}) := p_{\mathrm{tr}}(\mathbf{x}|y = -1)$ are positive and negative-conditional densities of the training distribution, respectively, and $\pi_{\mathrm{tr}} := p_{\mathrm{tr}}(y = +1)$ is the positive class-prior. Although we assume that class-prior $\pi_{\mathrm{tr}}$ is known in this paper, it can be estimated from positive and unlabeled data [44, 63, 15]. In addition, we suppose that a set of unlabeled instances $X_{\mathrm{te}}$ drawn from the test distribution is also given:

$$X_{\mathrm{te}} = \{\mathbf{x}_{\mathrm{te},n}\}_{n=1}^{N_{\mathrm{te}}} \sim p_{\mathrm{te}}(\mathbf{x}) = \pi_{\mathrm{te}} p_{\mathrm{te}}^{\mathrm{P}}(\mathbf{x}) + (1 - \pi_{\mathrm{te}}) p_{\mathrm{te}}^{\mathrm{n}}(\mathbf{x}), \tag{6}$$

where $p_{\mathrm{te}}^{\mathrm{P}}(\mathbf{x}) := p_{\mathrm{te}}(\mathbf{x}|y = +1)$, $p_{\mathrm{te}}^{\mathrm{n}}(\mathbf{x}) := p_{\mathrm{te}}(\mathbf{x}|y = -1)$, and $\pi_{\mathrm{te}} := p_{\mathrm{te}}(y = +1)$. We assume the class-imbalance in both unlabeled data, i.e., $\pi_{\mathrm{tr}}, \pi_{\mathrm{te}} \ll 1$. As shown later, the proposed method does not need to know $\pi_{\mathrm{te}}$ to maximize the AUC[1], which is beneficial in practice.

We consider a situation of *the positive distribution shift* between the training and test distributions, where the negative-conditional density does not change but the positive-conditional one can vary,

$$p_{\mathrm{tr}}^{\mathrm{n}}(\mathbf{x}) = p_{\mathrm{te}}^{\mathrm{n}}(\mathbf{x}), \ \ p_{\mathrm{tr}}^{\mathrm{P}}(\mathbf{x}) \neq p_{\mathrm{te}}^{\mathrm{P}}(\mathbf{x}). \tag{7}$$

This situation often occurs in imbalanced classification problems as described in Section 1. Our aim is to learn score function $s : \mathbb{R}^D \to \mathbb{R}$ that can maximize the AUC on the test distribution by using $X_{\mathrm{tr}}^{\mathrm{P}} \cup X_{\mathrm{tr}} \cup X_{\mathrm{te}}$. For score function $s$, we can use any differentiable function such as linear classifiers, kernel-based classifiers, or neural networks. Note that we also discuss the negative distribution shift, i.e., $p_{\mathrm{tr}}^{\mathrm{n}}(\mathbf{x}) \neq p_{\mathrm{te}}^{\mathrm{n}}(\mathbf{x})$ and $p_{\mathrm{tr}}^{\mathrm{P}}(\mathbf{x}) = p_{\mathrm{te}}^{\mathrm{P}}(\mathbf{x})$, in Section D.

### 4.2 Positive Distribution Shift Adaptation

In this subsection, we explain how to maximize the AUC under the positive distribution shift. The objective function to be maximized is the following smoothed AUC on the test distribution,

$$\mathrm{AUC}_{\sigma}(s) = \mathbb{E}_{\mathbf{x}^{\mathrm{P}} \sim p_{\mathrm{te}}^{\mathrm{P}}(\mathbf{x})} \mathbb{E}_{\mathbf{x}^{\mathrm{n}} \sim p_{\mathrm{te}}^{\mathrm{n}}(\mathbf{x})} \left[ f(\mathbf{x}^{\mathrm{P}}, \mathbf{x}^{\mathrm{n}}) \right], \tag{8}$$

where we set $f(\mathbf{x}^{\mathrm{P}}, \mathbf{x}^{\mathrm{n}}) := \sigma(s(\mathbf{x}^{\mathrm{P}}) - s(\mathbf{x}^{\mathrm{n}}))$. Since this AUC depends on $p_{\mathrm{te}}^{\mathrm{P}}(\mathbf{x})$ and $p_{\mathrm{te}}^{\mathrm{n}}(\mathbf{x})$, it seems be impossible to calculate in our setting where neither positive nor negative data in the test distribution are given. However, we show that calculation is possible. First, from the definition of marginal density $p_{\mathrm{te}}(\mathbf{x})$ in Eq. (6), the positive-conditional test density $p_{\mathrm{te}}^{\mathrm{P}}(\mathbf{x})$ can be expressed as

$$p_{\mathrm{te}}^{\mathrm{P}}(\mathbf{x}) = \frac{1}{\pi_{\mathrm{te}}} \left[ p_{\mathrm{te}}(\mathbf{x}) - (1 - \pi_{\mathrm{te}}) p_{\mathrm{te}}^{\mathrm{n}}(\mathbf{x}) \right], \tag{9}$$

where we assume that $\pi_{\mathrm{te}} > 0$. By substituting Eq. (9) into Eq. (8), we can obtain

$$\mathrm{AUC}_{\sigma}(s) = \frac{1}{\pi_{\mathrm{te}}} \left[ \mathbb{E}_{\mathbf{x} \sim p_{\mathrm{te}}(\mathbf{x})} \mathbb{E}_{\mathbf{x}^{\mathrm{n}} \sim p_{\mathrm{te}}^{\mathrm{n}}(\mathbf{x})} \left[ f(\mathbf{x}, \mathbf{x}^{\mathrm{n}}) \right] - (1 - \pi_{\mathrm{te}}) \mathbb{E}_{\mathbf{x}^{\mathrm{n}} \sim p_{\mathrm{te}}^{\mathrm{n}}(\mathbf{x})} \mathbb{E}_{\bar{\mathbf{x}}^{\mathrm{n}} \sim p_{\mathrm{te}}^{\mathrm{n}}(\mathbf{x})} \left[ f(\mathbf{x}^{\mathrm{n}}, \bar{\mathbf{x}}^{\mathrm{n}}) \right] \right]. \tag{10}$$

---

[1] To be precise, the specific value of $\pi_{\mathrm{te}}$ is not required, but we require that $\pi_{\mathrm{te}} \neq 0$, which will be discussed later.

---

**Algorithm 1** Training procedure of the proposed method

---

**Require:** Positive and unlabeled data in the training distribution $X_{\mathrm{tr}}^{\mathrm{P}} \cup X_{\mathrm{tr}}$, unlabeled data in the test distribution $X_{\mathrm{te}}$, positive class-prior in the training distribution $\pi_{\mathrm{tr}}$, mini-batch size $M$, and positive mini-batch size $P$

**Ensure:** Model parameters of score function $s$

 1: **repeat**
 2:     Sample positive data with size $P$ form $X_{\mathrm{tr}}^{\mathrm{P}}$
 3:     Sample unlabeled data with size $M - P$ from $X_{\mathrm{tr}} \cup X_{\mathrm{te}}$
 4:     Calculate the loss in Eq. (14) on the sampled positive and unlabeled data with $\pi_{\mathrm{tr}}$
 5:     Update model parameters of score function $s$ with the gradient of the loss
 6: **until** End condition is satisfied;

---

Here, the second term in Eq. (10) becomes constant $(1 - \pi_{\mathrm{te}})\mathbb{E}_{\mathbf{x}^{\mathrm{n}} \sim p_{\mathrm{te}}^{\mathrm{n}}(\mathbf{x})}\mathbb{E}_{\bar{\mathbf{x}}^{\mathrm{n}} \sim p_{\mathrm{te}}^{\mathrm{n}}(\mathbf{x})}\left[f(\mathbf{x}^{\mathrm{n}}, \bar{\mathbf{x}}^{\mathrm{n}})\right] = (1 - \pi_{\mathrm{te}})/2$ because $\sigma(z) + \sigma(-z) = 1$ for all $z \in \mathbb{R}$ [58, 6, 59]. Therefore, we can ignore the second term in Eq. (10) to learn score function $s$. Next, by using the assumption of the positive distribution shift, $p_{\mathrm{tr}}^{\mathrm{n}}(\mathbf{x}) = p_{\mathrm{te}}^{\mathrm{n}}(\mathbf{x})$, and the definition of the marginal density $p_{\mathrm{tr}}(\mathbf{x})$ in Eq. (5), we can express the negative-conditional test density $p_{\mathrm{te}}^{\mathrm{n}}(\mathbf{x})$ as

$$p_{\mathrm{te}}^{\mathrm{n}}(\mathbf{x}) = p_{\mathrm{tr}}^{\mathrm{n}}(\mathbf{x}) = \frac{1}{1 - \pi_{\mathrm{tr}}}\left[p_{\mathrm{tr}}(\mathbf{x}) - \pi_{\mathrm{tr}}p_{\mathrm{tr}}^{\mathrm{P}}(\mathbf{x})\right], \tag{11}$$

where we assume that $\pi_{\mathrm{tr}} < 1$. As before, by substituting Eq. (11) into the first term in Eq. (10), we can obtain

$$\mathrm{AUC}_\sigma(s) = \frac{1}{\pi_{\mathrm{te}}(1 - \pi_{\mathrm{tr}})}\left[\mathbb{E}_{\mathbf{x} \sim p_{\mathrm{te}}(\mathbf{x})}\mathbb{E}_{\bar{\mathbf{x}} \sim p_{\mathrm{tr}}(\mathbf{x})}\left[f(\mathbf{x}, \bar{\mathbf{x}})\right] - \pi_{\mathrm{tr}}\mathbb{E}_{\mathbf{x} \sim p_{\mathrm{te}}(\mathbf{x})}\mathbb{E}_{\mathbf{x}^{\mathrm{P}} \sim p_{\mathrm{tr}}^{\mathrm{P}}(\mathbf{x})}\left[f(\mathbf{x}, \mathbf{x}^{\mathrm{P}})\right]\right] + C, \tag{12}$$

where $C$ represents constant terms that do not depend on score function $s$. Since coefficient $1/(\pi_{\mathrm{te}}(1 - \pi_{\mathrm{tr}}))$ does not affect the optimization for $s$, our loss function to be minimized is as follows,

$$\mathcal{L}(s) := -\mathbb{E}_{\mathbf{x} \sim p_{\mathrm{te}}(\mathbf{x})}\mathbb{E}_{\bar{\mathbf{x}} \sim p_{\mathrm{tr}}(\mathbf{x})}\left[f(\mathbf{x}, \bar{\mathbf{x}})\right] + \pi_{\mathrm{tr}}\mathbb{E}_{\mathbf{x} \sim p_{\mathrm{te}}(\mathbf{x})}\mathbb{E}_{\mathbf{x}^{\mathrm{P}} \sim p_{\mathrm{tr}}^{\mathrm{P}}(\mathbf{x})}\left[f(\mathbf{x}, \mathbf{x}^{\mathrm{P}})\right]. \tag{13}$$

This loss depends on the positive and marginal training densities, marginal test densities, and class-prior. Therefore, we can approximate this loss with given data $X_{\mathrm{tr}}^{\mathrm{P}} \cup X_{\mathrm{tr}} \cup X_{\mathrm{te}}$ and $\pi_{\mathrm{tr}}$. Specifically, the empirical estimate of the loss function is given as follows,

$$\hat{\mathcal{L}}(s) = -\frac{1}{N_{\mathrm{te}}N_{\mathrm{tr}}}\sum_{n,m=1}^{N_{\mathrm{te}},N_{\mathrm{tr}}} f(\mathbf{x}_{\mathrm{te},n}, \mathbf{x}_{\mathrm{tr},m}) + \frac{\pi_{\mathrm{tr}}}{N_{\mathrm{te}}N_{\mathrm{tr}}^{\mathrm{P}}}\sum_{n,m=1}^{N_{\mathrm{te}},N_{\mathrm{tr}}^{\mathrm{P}}} f(\mathbf{x}_{\mathrm{te},n}, \mathbf{x}_{\mathrm{tr},m}^{\mathrm{P}}). \tag{14}$$

Algorithm 1 shows the pseudocode of our training procedure with stochastic gradient methods. Note that although we have used the sigmoid function to represent the AUC in Eq. (2), as long as we use symmetric functions (i.e., function $\sigma$ satisfying $\sigma(z) + \sigma(-z) = K$ for any $z \in \mathbb{R}$ and $K$ is a constant [6]), we can derive the loss function of the same form in Eq. (13). The symmetric functions include a wide range of functions such as sigmoid, ramp, and unhinged functions [6]. In addition, the loss corrections for the proposed method such as the non-negative correction [25], which is commonly used in ordinary PU learning to avoid overfitting, are discussed in detail in Section B.

### 4.3 Discussion about Class-priors

In the process of deriving our loss function, we assume $\pi_{\mathrm{tr}} < 1$ and $\pi_{\mathrm{te}} > 0$. We discuss why this assumption is necessary in our setting. When $\pi_{\mathrm{te}} = 0$, all unlabeled data $X_{\mathrm{te}}$ become negative data. In this case, our given datasets $X_{\mathrm{tr}}^{\mathrm{P}} \cup X_{\mathrm{tr}} \cup X_{\mathrm{te}}$ do not have any information about positive data in the test distribution since $p_{\mathrm{tr}}^{\mathrm{P}}(\mathbf{x}) \neq p_{\mathrm{te}}^{\mathrm{P}}(\mathbf{x})$. Therefore, we cannot calculate the AUC on the test distribution. However, in this case, we can easily create trivial optimal classifiers that classify all data as negative. Thus, this is not an issue.

When $\pi_{\mathrm{tr}} = 1$, all unlabeled data $X_{\mathrm{tr}}$ become positive data on the training distribution. $X_{\mathrm{tr}}^{\mathrm{P}} \cup X_{\mathrm{tr}}$ (positive data in the training distribution) have no information to extract positive and negative data from $X_{\mathrm{te}}$ since $p_{\mathrm{tr}}^{\mathrm{P}}(\mathbf{x}) \neq p_{\mathrm{te}}^{\mathrm{P}}(\mathbf{x})$. Thus, in this case, we also cannot calculate the AUC on the test distribution. However, it is extremely rare for all unlabeled data to be positive in imbalanced data problems where the ratio of positive data in unlabeled data is low. Thus, this is also not a problem in practice.

## 4.4 Extension

In some applications, a small number of labeled negative data as well as labeled positive data might be available from the training distribution. The proposed method can be easily extended to such cases. Specifically, by Eq. (10), the AUC on the test distribution without constant terms is represented as

$$\mathrm{AUC}_{\sigma}(s) \propto \mathbb{E}_{\mathbf{x} \sim p_{\mathrm{te}}(\mathbf{x})} \mathbb{E}_{\mathbf{x}^{\mathrm{n}} \sim p_{\mathrm{te}}^{\mathrm{n}}(\mathbf{x})} \left[ f(\mathbf{x}, \mathbf{x}^{\mathrm{n}}) \right] = \mathbb{E}_{\mathbf{x} \sim p_{\mathrm{te}}(\mathbf{x})} \mathbb{E}_{\mathbf{x}^{\mathrm{n}} \sim p_{\mathrm{tr}}^{\mathrm{n}}(\mathbf{x})} \left[ f(\mathbf{x}, \mathbf{x}^{\mathrm{n}}) \right] =: \mathcal{R}(s), \quad (15)$$

where we used assumption $p_{\mathrm{tr}}^{\mathrm{n}}(\mathbf{x}) = p_{\mathrm{te}}^{\mathrm{n}}(\mathbf{x})$. Loss $\mathcal{R}(s)$ can be approximated with test unlabeled data and labeled negative data. Thus, the following modified loss function to be minimized can be used:

$$\mathcal{L}_{\mathrm{modified}}(s) := \alpha \mathcal{L}(s) - (1 - \alpha) \mathcal{R}(s), \quad (16)$$

where $\alpha \in [0, 1]$ is a weighting hyperparameter.

## 5 Experiments

In this section, we empirically demonstrate the effectiveness of the proposed method under the positive distribution shift with real-world datasets.

### 5.1 Data

We utilized four widely used real-world datasets in the main paper: MNIST [30], FashionMNIST [57], SVHN [40], and CIFAR10 [26]. MNIST consists of hand-written images of 10 digits. Each image is represented by gray-scale with $28 \times 28$ pixels. FashionMNIST consists of images of 10 fashion categories where each image is represented by gray scale with $28 \times 28$ pixels. SVHN consists of $32 \times 32$ RGB images of printed 10 digits clipped from photographs of house number plates. We converted SVHN into gray-scale for simplicity. CIFAR10 consists of $32 \times 32$ RGB images of 10 animal and vehicle categories. In Section E.3, we also evaluated the proposed method with two tabular datasets with distribution shifts (HReadmission and Hypertension) [14].

For MNIST and SVHN, we used even digits as the negative class and odd digits as the positive class. Following the previous studies [16, 18], data with digits '7' and '9' were used as positive data appearing in the training distribution and data with digits '1', '3', '5', '7', and '9' were used as positive data appearing in the test distribution. This simulates a situation where new types of positive data, which did not appear during the training, appear in the test environment. For FashionMNIST, following the study [59], we used upper garments ('T-shift', 'Pullover', 'Dress', 'Coat', and 'Shift') as the negative class and the others as the positive class. Positive data with the 'Trouser' and 'Bag' appeared in the training distribution and those with the 'Trouser', 'Bag', 'Sandal', 'Sneaker', and 'Ankle boot' appeared in the test distribution. For CIFAR10, we used the animal categories as the negative class and the vehicles as the positive class. Positive data with the 'Airplane' appeared in the training distribution and those with the 'Airplane', 'Automobile', 'Ship', and 'Truck' appeared in the test distribution as in the previous study [16].

For each dataset, we used 10 positive and $5,000$ unlabeled data in the training distribution and $5,000$ unlabeled data in the test distribution for training. In addition, we used 5 positive and 500 unlabeled data in the training distribution and 500 unlabeled data in the test distribution for validation. We used $1,500$ positive and $1,500$ negative data in the test distribution as test data for evaluation. There is no overlap between training, validation, and test datasets. The positive class-prior on the training distribution $\pi_{\mathrm{tr}}$ was set to $0.1$ and that on the training distribution $\pi_{\mathrm{te}}$ was changed within $\{0.1, 0.2, 0.3\}$. For each case of the positive class-prior pairs, we conducted eight experiments while changing the random seeds and evaluated mean test AUC.

### 5.2 Comparison Methods

We compared the proposed method (Ours) with nine neural network-based comparison methods: CE, nnPU [25], puAUC [58], PULA [21], AnnPU, ApuAUC, APULA, CpuAUC, and PURR [16].

CE, nnPU, puAUC, and PULA use positive and unlabeled data in the training distribution for learning classifiers. These methods do not adapt to the test distribution. CE learns neural network parameters by minimizing the cross-entropy loss. This method naively treats unlabeled data as negative data. nnPU is a widely used PU learning method with a non-negative risk estimator. This method rewrites

the classification risk with positive and unlabeled data and minimizes the rewritten non-negative classification risk to learn neural network parameters. PULA is a recent PU learning method that learns a neural network classifier with the label distribution alignment. nnPU and PULA used the class-prior in the training distribution as in the proposed method. puAUC learns a neural network classifier by maximizing the AUC that is calculated from positive and unlabeled data.

AnnPU, ApuAUC, APULA, CpuAUC, and PURR use positive and unlabeled data in the training distribution and unlabeled data in the test distribution. AnnPU, ApuAUC, and APULA use unlabeled data in both the training and test distributions instead of those in the training distribution of nnPU, puAUC, and PULA respectively. AnnPU and APULA used the composite class-prior on the combined unlabeled data (i.e., $\pi = \frac{\pi_{\mathrm{tr}} N_{\mathrm{tr}} + \pi_{\mathrm{te}} N_{\mathrm{te}}}{N_{\mathrm{tr}} + N_{\mathrm{te}}}$). In our preliminary experiments, the methods that use unlabeled data from the test distribution instead of data from both distributions were also evaluated. However, they performed worse than AnnPU, ApuAUC, and APULA. Thus, we omitted them. CpuAUC learns the invariant feature representations to mitigate the discrepancy of training and test distributions. Specifically, this method minimizes the negative AUC loss used in puAUC and the CORAL loss, which is a widely used in domain adaptation studies to match the second-order statistics of both distributions [53, 52]. PURR is a PU learning method for the positive distribution shift. This method rewrites the classification risk assuming the negative-conditional density does not change. To rewrite the risk, PURR requires the class-prior in the test distribution to be available, whereas the proposed method does not.

## 5.3 Settings

For MNIST, FashionMNIST, and SVHN, all methods used a feed-forward neural network with three hidden layers. The number of hidden nodes was 128 and the ReLU activation function was used for each hidden layer. For CIFAR10, all methods used a convolutional neural network, which consisted of two convolutional blocks followed by a feed-forward neural network with two hidden layers. The first (second) convolutional block comprised a 6 (16) filter $5 \times 5$ convolution, the ReLU activation function, and a $2 \times 2$ max-pooling layer [38]. The numbers of nodes in the two hidden layers were 120 and 84, and the ReLU activation function was used. For PULA and APULA, margin $\rho$ was selected from $\{0.1, 1, 10\}$. For CpuAUC, the CORAL loss was applied to the last hidden layer and its regularization weight was selected within $\{100, 10, 1, 0.1, 0.01\}$. For PULA, APULA, and CpuAUC, the best test results were reported. For all methods, we used the Adam optimizer [24] with a learning rate of $10^{-4}$. We set a mini-batch size $M$ to 512, a positive mini-batch size $P$ to 10, and the maximum number of epochs to 200. The loss on validation data was used for early stopping to avoid overfitting. All methods were implemented using Pytorch [43] and all experiments were conducted on a Linux server with an Intel Xeon CPU and A100 GPU.

## 5.4 Results

We evaluated the performance of the proposed method under the positive distribution shift with imbalanced datasets. Table 1 shows the average test AUCs of each method on the four datasets (their standard deviations are reported in Section E.4). The proposed method performed the best or comparably to it in almost all cases (11 out of 12 cases). The non-adaptation methods (CE, nnPU, puAUC, and PULA) that do not use data in the test distribution performed worse than the proposed method in many cases. This result indicates that adaptation using data in the test distribution is essential when there are distribution shifts. Although AnnPU, ApuAUC, and APULA used unlabeled data in the test distribution, they also did not work well. This is because their loss functions that naively use unlabeled data in the test distribution are not designed to maximize the performance on the test distribution. CpuAUC that learns invariant feature representations to mitigate the distribution gap tended to perform worse than the proposed method since the invariant features are not theoretically guaranteed to maximize the performance under the positive distribution shift. PURR performed well with MNIST and FashionMNIST since it is designed to adapt to the positive distribution shift and both datasets are relatively simple data where training is easy even in imbalanced data settings. We note that PURR used information on true class-priors on the test distribution, which is not used in the proposed method. By maximizing the AUC, the proposed method outperformed PURR with SVHN and CIFAR10 by a large margin, which are more complex than MNIST and FashionMNIST. As for the results of each class-prior pair, the proposed method performed better as the positive class-prior on the test distribution $\pi_{\mathrm{te}}$ increased. Since unlabeled data $X_{\mathrm{te}}$ has many positive data

Table 1: Average test AUCs with different class-prior pairs. Values in bold are not statistically different at the $5\%$ level from the best performing method in each row according to a paired t-test. '# best' row represents the number of results of each method that are the best or comparable to it.

| Data | $\pi_{\text{te}}$ | Ours | CE | nnPU | puAUC | PULA | AnnPU | ApuAUC | APULA | CpuAUC | PURR |
|------|------|------|----|------|-------|------|-------|--------|-------|--------|------|
| MNIST | 0.1 | **0.723** | **0.796** | **0.782** | **0.801** | **0.802** | **0.782** | **0.798** | **0.793** | **0.804** | **0.815** |
| | 0.2 | **0.854** | 0.796 | 0.784 | 0.801 | 0.802 | 0.785 | 0.793 | 0.769 | 0.803 | **0.871** |
| | 0.3 | **0.902** | 0.796 | 0.783 | 0.800 | 0.802 | 0.780 | 0.786 | 0.745 | 0.802 | **0.914** |
| Fashion | 0.1 | 0.787 | 0.932 | 0.825 | 0.937 | **0.954** | 0.772 | 0.920 | **0.939** | 0.943 | 0.920 |
| MNIST | 0.2 | **0.891** | 0.930 | 0.825 | 0.937 | **0.955** | 0.771 | 0.890 | **0.911** | 0.943 | **0.929** |
| | 0.3 | **0.960** | 0.930 | 0.825 | 0.938 | **0.955** | 0.866 | 0.870 | 0.863 | **0.944** | **0.961** |
| SVHN | 0.1 | **0.554** | 0.504 | 0.501 | 0.518 | 0.494 | 0.501 | 0.512 | 0.492 | 0.524 | 0.511 |
| | 0.2 | **0.660** | 0.503 | 0.501 | 0.518 | 0.494 | 0.501 | 0.507 | 0.490 | 0.523 | 0.522 |
| | 0.3 | **0.736** | 0.503 | 0.501 | 0.518 | 0.494 | 0.501 | 0.507 | 0.490 | 0.523 | 0.519 |
| CIFAR10 | 0.1 | **0.727** | **0.682** | 0.455 | **0.749** | **0.751** | 0.489 | **0.739** | **0.750** | **0.750** | 0.434 |
| | 0.2 | **0.825** | 0.679 | 0.467 | 0.741 | 0.739 | 0.491 | 0.732 | 0.742 | 0.744 | 0.536 |
| | 0.3 | **0.874** | 0.682 | 0.451 | 0.750 | 0.738 | 0.545 | 0.722 | 0.750 | 0.749 | 0.710 |
| # best | | 11 | 2 | 1 | 3 | 5 | 1 | 2 | 4 | 3 | 5 |

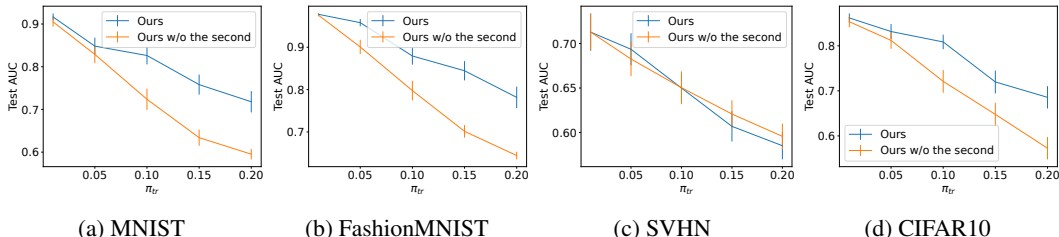

|  (a) MNIST | (b) FashionMNIST | (c) SVHN | (d) CIFAR10 |

Figure 2: Ablation study of our loss function: average test AUCs and standard errors over different class-priors on the test distribution $\pi_{\text{te}}$ within $\{0.1, 0.2, 0.3\}$ when changing the class-prior on the training distribution $\pi_{\text{tr}}$. Ours w/o the second is Ours without the second term in Eq. (13).

when $\pi_{\text{te}}$ is large, the proposed method more easily extracted information on positive data from $X_{\text{te}}$ and consequently performed well.

We conducted an ablation study of the proposed method. When positive class-prior on the training distribution $\pi_{\text{tr}}$ is small, all unlabeled data $X_{\text{tr}}$ might be treated as negative data. In this case, existing methods can be directly applied to maximize the AUC on the test distribution by using unlabeled data in the test distribution $X_{\text{te}}$ and (pseudo) negative data $X_{\text{tr}}$ [45, 58, 6, 59]. This approach is equivalent to the proposed method without the second term (or $\pi_{\text{tr}} = 0$) in Eq. (13), denoted as Ours w/o the second. Figure 2 shows the average test AUCs and the standard errors of the proposed method (Ours) and Ours w/o the second when changing the value of class-prior $\pi_{\text{tr}}$ within $\{0.01, 0.05, 0.1, 0.15, 0.2\}$. Ours consistently outperformed Ours w/o the second across all class-priors $\pi_{\text{tr}}$ except for SVHN. As class-prior $\pi_{\text{tr}}$ is increased, the difference between the two methods tended to become larger. This is because the assumption of Ours w/o the second (i.e., unlabeled data $X_{\text{tr}}$ are all negative data) is significantly broken when $\pi_{\text{tr}}$ is large. With SVHN, the performances of both methods did not differ. This result suggests that it is acceptable to consider all unlabeled data as negative data in some datasets. However, overall, these results show that the naive approximation is generally ineffective and our theoretically grounded loss function is important even with small $\pi_{\text{tr}}$.

We evaluated the proposed method with estimated class-priors on the training distribution since class-prior information might be unavailable in practice. To estimate the class-prior from positive and unlabeled training data, we used the kernel embedding-based class-prior estimation method [44]. Table 2 shows the results. Here, Ours w/ true and Ours w/ est are the proposed method using true class-prior $\pi_{\text{tr}}$ and estimated class-prior $\pi_{\text{tr}}^{\text{est}}$, respectively. We found that true class-prior $\pi_{\text{tr}} = 0.1$ was accurately estimated from positive and unlabeled data in the training distribution. As a result, Ours w/

Table 2: Results with estimated class-priors on the training distribution $\pi_{\text{tr}}^{\text{est}}$: average test AUCs [%] over different class-priors on the test distribution when training class-prior $\pi_{\text{tr}}$ is 0.1. FMNIST is an acronym for FashionMNIST.

| Data | $\pi_{\text{tr}}^{\text{est}}$ | Ours w/ true | Ours w/ est |
|------|------|------|------|
| MNIST | 0.117 | 0.826 | 0.845 |
| FMNIST | 0.077 | 0.879 | 0.857 |
| SVHN | 0.078 | 0.650 | 0.652 |
| CIFAR10 | 0.146 | 0.809 | 0.811 |

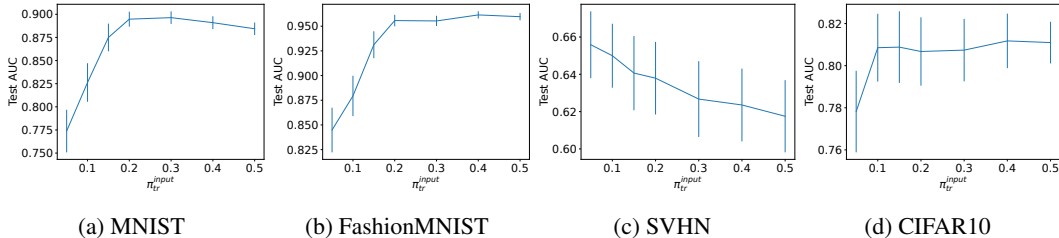

| (a) MNIST | (b) FashionMNIST | (c) SVHN | (d) CIFAR10 |

Figure 3: Results in the case where true and input class-priors on the training distribution $\pi_{\mathrm{tr}}$ can be different: average test AUCs and standard errors over different class-priors on the test distribution $\pi_{\mathrm{te}}$ within $\{0.1, 0.2, 0.3\}$ with true training class-prior $\pi_{\mathrm{tr}} = 0.1$ when changing the input class-prior on the training distribution.

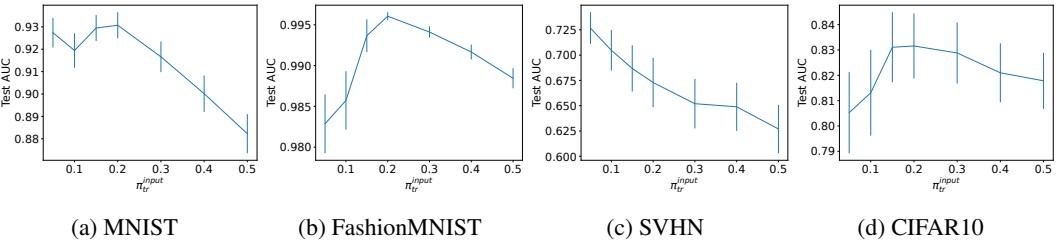

| (a) MNIST | (b) FashionMNIST | (c) SVHN | (d) CIFAR10 |

Figure 4: Results in the case where true and input class-priors on the training distribution $\pi_{\mathrm{tr}}$ can be different when the positive distribution drastically changed: average test AUCs and standard errors with true training class-prior $\pi_{\mathrm{tr}} = 0.1$ when changing the input class-prior on the training distribution.

est and Ours w/ true mostly performed similarly. This result indicates that the proposed method works well even when class-prior information is unavailable in the training distribution, which is preferable in practice. Additionally, we investigated how the performance of the proposed method changes when there is a difference between true and input class-priors on the training distribution $\pi_{\mathrm{tr}}$. Here, the input class-prior is used in the loss function of Eq. (13) and the true class-prior is used for data generation. Figure 3 shows the average test AUCs and the standard errors of the proposed method (Ours) when changing the input class-priors. Interestingly, the performance tended to improve when values larger than the true class-prior (0.1) were inputted in most datasets. One reason for this result is that since there was some overlap between the types of positive data in the test and training distributions, strengthening the influence on labeled positive data by increasing the input class-prior in Eq. (13) improved the performance. To confirm this, we conducted additional experiments by excluding the types of positive data from the test distribution that were used in the training distribution (e.g., digits '7' and '9' were excluded from the test distribution in MNIST). Figure 4 shows the results. As expected, the performance did not improve even with large input class-priors.

We investigated how the performance of the proposed method changes when the number of labeled positive data $N_{\mathrm{tr}}^{\mathrm{p}}$ is increased. Figure 5 shows the average test AUCs and the standard errors with different numbers of labeled positive data. As expected, the performance of the proposed method tended to improve as $N_{\mathrm{tr}}^{\mathrm{p}}$ increased. By using much information on labeled positive data in the training distribution, the proposed method can accurately estimate the AUC on the test distribution.

We investigated the performance of the proposed method when changing the number of unlabeled data in the test distribution in Table 3. As expected, the performance of the proposed method tended to increase as the number of unlabeled data in the test distribution $N_{\mathrm{te}}$ increased. The proposed method tended to outperform puAUC, which does not use unlabeled data in the test distribution, even when $N_{\mathrm{te}} = 500$. Since many unlabeled data are often easy to collect, the proposed method is useful in practice.

We evaluated the modified loss function in Eq. (16) for the cases where positive, negative, and unlabeled data in the training distribution and unlabeled data in the test distribution are available. Table 4 shows the results. Here, the number of labeled negative data was set to 20 and other settings

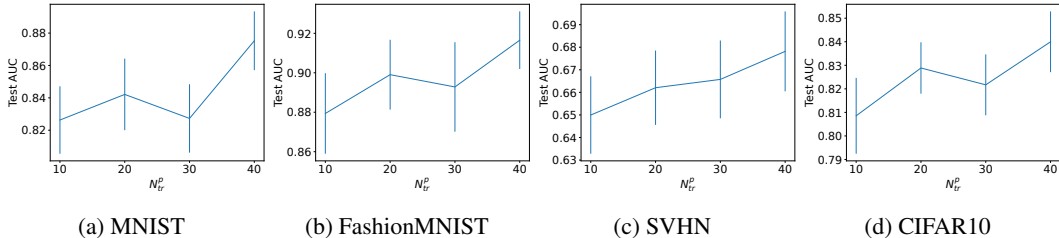

| | (a) MNIST | (b) FashionMNIST | (c) SVHN | (d) CIFAR10 |

Figure 5: Average test AUCs and standard errors of the proposed method over different class-priors on the test distribution when changing the number of labeled positive data in the training distribution $N_{\mathrm{tr}}^{\mathrm{p}}$.

Table 3: Results of the proposed method with different numbers of unlabeled data in the test distribution : average test AUCs over different class-prior pairs $\pi_{\mathrm{te}}$ in the test distribution within $\{0.1, 0.2, 0.3\}$ when training class-prior $\pi_{\mathrm{tr}}$ is 0.1. 'Base' represents the result of puAUC that does not use unlabeled data in the test distribution.

| $N_{\mathrm{te}}$ | 100 | 500 | 1000 | 2000 | 5000 | Base |
|---|---|---|---|---|---|---|
| MNIST | 0.773 | 0.813 | 0.823 | 0.826 | 0.827 | 0.801 |
| Fashion MNIST | 0.863 | 0.913 | 0.918 | 0.911 | 0.879 | 0.938 |
| SVHN | 0.516 | 0.579 | 0.616 | 0.626 | 0.650 | 0.518 |
| CIFAR10 | 0.683 | 0.757 | 0.770 | 0.778 | 0.809 | 0.746 |

Table 4: Results when a few labeled negative data are available on the training distribution: average test AUCs for each test class-prior. FMNIST is an acronym for FashionMNIST.

| Data | MNIST | | | FMNIST | | | SVHN | | | CIFAR10 | | |
|---|---|---|---|---|---|---|---|---|---|---|---|---|
| $\pi_{\mathrm{te}}$ | 0.1 | 0.2 | 0.3 | 0.1 | 0.2 | 0.3 | 0.1 | 0.2 | 0.3 | 0.1 | 0.2 | 0.3 |
| $\alpha = 1.0$ | 0.723 | 0.854 | 0.902 | 0.787 | 0.891 | 0.960 | 0.554 | 0.660 | 0.736 | 0.727 | 0.825 | 0.873 |
| $\alpha = 0.999$ | 0.727 | 0.875 | 0.904 | 0.816 | 0.939 | 0.947 | 0.542 | 0.662 | 0.743 | 0.731 | 0.838 | 0.870 |
| $\alpha = 0.0$ | 0.517 | 0.577 | 0.661 | 0.732 | 0.794 | 0.838 | 0.503 | 0.506 | 0.510 | 0.602 | 0.594 | 0.697 |

are the same as those in the previous experiments. First, the proposed method with $\alpha = 0$, which maximizes the AUC on the test distribution in Eq. (15) with labeled negative and test unlabeled data, did not work well. This would be because the number of positive data in test unlabeled data was small, and thus it was difficult to extract information on such positive data from unlabeled data by using a few labeled negative data only. The proposed method with $\alpha = 0.999$, which uses positive, negative, and unlabeled data in the training distribution and unlabeled data in the test distribution, slightly tended to perform better than that with $\alpha = 1.0$, which is equivalent to the proposed method described in Section 4.2. This result suggests that using a few labeled negative data is useful in our framework.

## 6 Conclusion

In this paper, we proposed a AUC maximization method under the positive distribution shift. We theoretically showed the AUC on the test distribution can be maximized by using positive and unlabeled data in the training distribution and unlabeled data in the test distribution when assuming the positive distribution shift. The derived AUC has the advantage of being simple and easy to implement. The experiments with six real-world datasets demonstrated that the proposed method outperformed existing methods under the positive distribution shift with imbalanced data.

## 7 Limitations

The proposed method performs well in the positive distribution shift. As stated in Section 1, we can expect that the positive distribution shift must occur in many real-world applications. However, the proposed method is not guaranteed to keep working well if different distribution shifts occur unexpectedly (i.e., the negative-conditional density changes). To deal with this problem, methods should be developed that can predict the types of the shift from given data.

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

# A Impact Statements

Although the proposed method performed well, there is a possibility of misclassification in practice. In particular, the misclassification can lead to serious incidents in cases such as cyber/physical security and medical care, which are typical examples of imbalanced data. Therefore, this method should be used as a support tool for humans to make a final decision.

# B Proposed Method with Loss Corrections

In this section, we discuss the loss (risk) corrections commonly used in ordinary PU learning to mitigate overfitting [25, 16]. Specifically, the loss correction is used to prevent empirical estimates of the risk from taking negative values even when the risk never takes negative, which often leads to overfitting.

First, we provide a summary of our results: (1) The non-negative loss correction, which is widely used for ordinary PU learning [25, 16], is not effective in our AUC maximization framework. This is because zero is not a tight lower bound of our expected loss. (2) When class-prior in the test distribution $\pi_{\mathrm{te}}$ is available, we can derive a tighter lower bound. By using this for the loss correction, we can often enhance the performance of the proposed method. Below, we describe the details.

**Non-negative loss correction.** In this section, for clarity, we consider minimizing the AUC risk,

$$R_\sigma(s) := \mathbb{E}_{\mathbf{x}^{\mathrm{p}} \sim p_{\mathrm{te}}^{\mathrm{P}}(\mathbf{x})} \mathbb{E}_{\mathbf{x}^{\mathrm{n}} \sim p_{\mathrm{te}}^{\mathrm{n}}(\mathbf{x})}[f(\mathbf{x}^{\mathrm{n}}, \mathbf{x}^{\mathrm{P}})], \tag{17}$$

which is equivalent to maximize the AUC in Eq. (8) since $\mathrm{AUC}_\sigma(s) = 1 - R_\sigma(s)$. The minimum value of this risk is zero. Then, the corresponding loss for Eq. (13) becomes

$$\mathcal{L}_{\mathrm{risk}}(s) := \mathbb{E}_{\mathbf{x} \sim p_{\mathrm{te}}(\mathbf{x})} \mathbb{E}_{\bar{\mathbf{x}} \sim p_{\mathrm{tr}}(\mathbf{x})}[f(\bar{\mathbf{x}}, \mathbf{x})] - \pi_{\mathrm{tr}} \mathbb{E}_{\mathbf{x} \sim p_{\mathrm{te}}(\mathbf{x})} \mathbb{E}_{\mathbf{x}^{\mathrm{P}} \sim p_{\mathrm{tr}}^{\mathrm{P}}(\mathbf{x})}[f(\mathbf{x}^{\mathrm{P}}, \mathbf{x})]. \tag{18}$$

This loss is derived from the AUC risk with unlabeled and negative test densities, i.e.,

$$\mathcal{L}_{\mathrm{risk}}(s) = (1 - \pi_{\mathrm{tr}}) \mathbb{E}_{\mathbf{x} \sim p_{\mathrm{te}}(\mathbf{x})} \mathbb{E}_{\mathbf{x}^{\mathrm{n}} \sim p_{\mathrm{te}}^{\mathrm{n}}(\mathbf{x})}[f(\mathbf{x}^{\mathrm{n}}, \mathbf{x})]. \tag{19}$$

Since the AUC risk does not take negative values, the loss should not also take negative values. Therefore, to prevent negative values of its empirical estimate $\hat{\mathcal{L}}_{\mathrm{risk}}(s)$, we can use the empirical loss with the absolute value function $|\hat{\mathcal{L}}_{\mathrm{risk}}(s)|$ for the optimization. This type of correction is successfully used in PU learning [16] or other weakly supervised learning [34]. Table 5 shows the results of the proposed method with $|\hat{\mathcal{L}}_{\mathrm{risk}}(s)|$ (Ours w/ nn). The results of Ours w/ nn and the proposed method without the loss correction (Ours) were almost identical and thus the non-negative loss correction was not effective in our framework unlike ordinary PU learning studies [25].

The reason of the ineffectiveness of Ours w/ nn is that the non-negative constraint is insufficient/weak in our loss (Eq. (18)). In fact, since $p_{\mathrm{te}}^{\mathrm{n}}(\mathbf{x})$ is contained in $p_{\mathrm{te}}(\mathbf{x})(= \pi_{\mathrm{te}} p_{\mathrm{te}}^{\mathrm{P}}(\mathbf{x}) + (1 - \pi_{\mathrm{te}}) p_{\mathrm{te}}^{\mathrm{n}}(\mathbf{x}))$, the minimum value of the expected loss $\mathcal{L}_{\mathrm{risk}}(s)$ in Eq. (19) actually be greater than zero. Thus, the empirical loss $\hat{\mathcal{L}}_{\mathrm{risk}}(s)$ could not be sufficiently constrained with the non-negativity; the performance of Ours w/ nn did not improve.

Note that the non-negative correction is effective in ordinary PU learning studies [25, 16]. This is because the minimum value of the risk $R_{\mathrm{n}}^-(g) := \mathbb{E}_{\mathbf{x} \sim p^{\mathrm{n}}(\mathbf{x})}[\ell(g(\mathbf{x}), -1)]$ used in the study [25] is zero, where $\ell$ is a point-wise loss and $g$ is a decision function. This creates a gap in the effectiveness of the non-negative correction in our framework and the previous study [25].

**Loss correction with class-prior in the test distribution $\pi_{\mathrm{te}}$.** If class-prior $\pi_{\mathrm{te}}$ is known, we can derive a tighter lower bound of our loss $\mathcal{L}_{\mathrm{risk}}(s)$. Specifically, we can obtain

$$\mathcal{L}_{\mathrm{risk}}(s) = (1 - \pi_{\mathrm{tr}}) \mathbb{E}_{\mathbf{x} \sim p_{\mathrm{te}}(\mathbf{x})} \mathbb{E}_{\mathbf{x}^{\mathrm{n}} \sim p_{\mathrm{te}}^{\mathrm{n}}(\mathbf{x})}[f(\mathbf{x}^{\mathrm{n}}, \mathbf{x})] \geq \frac{(1 - \pi_{\mathrm{tr}})(1 - \pi_{\mathrm{te}})}{2} =: b > 0. \tag{20}$$

Here, we used $p_{\mathrm{te}}(\mathbf{x}) = \pi_{\mathrm{te}} p_{\mathrm{te}}^{\mathrm{P}}(\mathbf{x}) + (1 - \pi_{\mathrm{te}}) p_{\mathrm{te}}^{\mathrm{n}}(\mathbf{x})$ and the fact that the AUC risk between the same densities is $1/2$ as described in Section 4.2. As a result, we can use $|\hat{\mathcal{L}}_{\mathrm{risk}}(s) - b| + b$ for the optimization. Our method with this correction (Our w/ b) tended to enhance the performance of it without the correction (Ours) in Table 5. However, note that Ours has the strong advantage of not requiring the class-prior and performed better than existing methods.

Table 5: Comparison with the proposed methods that use the loss corrections. Ours w/ nn used the non-negative loss correction and Ours w/ b uses $b = (1 - \pi_{\text{te}})(1 - \pi_{\text{tr}})/2$ for the correction: average test AUCs over different class-prior pairs $\pi_{\text{te}}$ in the test distribution within $\{0.1, 0.2, 0.3\}$ when training class-prior $\pi_{\text{tr}}$ is 0.1.

| Data | Ours | Ours w/ nn | Ours w/ b |
|------|------|-----------|-----------|
| MNIST | 0.827 | 0.816 | 0.885 |
| Fashion MNIST | 0.879 | 0.879 | 0.954 |
| SVHN | 0.650 | 0.650 | 0.572 |
| CIFAR10 | 0.809 | 0.792 | 0.829 |

**Dynamics of training loss, validation loss, and test AUC.** Lastly, we compared the dynamics of training loss, validation loss, and test AUC of these methods (Ours, Ours w/ nn, and Ours w/ b) in Figure 6. Here, the validation loss was calculated with validation PU data (i.e., PU data in the training distribution and U data in the test distribution). As learning progressed, the training losses of Ours and Ours w/ nn became smaller than the lower bound $b = 0.315$; the validation losses and test AUCs tended to stop improving or to become worse after the training losses were below $b = 0.315$. However, since the validation loss and test AUC were well correlated, Ours and Ours w/nn could select good models by using early-stopping with the validation loss. Ours w/ b tended to maintain good test AUCs without overfitting even when the learning is processed by evading the training loss to take smaller values than $b = 0.315$.

## C   How to Determine the Classification Threshold

The proposed method allows us to sort data in score order. In practical use, this is beneficial in many situations. For example, in anomaly detection, experts or operators can check data with high scores within the cost they can spend or until anomalous data do not appear. In disease diagnosis, patients with higher scores can be prioritized for detailed examination. In recommendation systems, products can be presented to users in order of score. However, a classification threshold may be required to classify positive and negative data in some situations. Below, we explain how to determine the classification threshold.

First, we have $N_{\text{te}}$ unlabeled data in the test distribution. When the true positive class-prior in the test distribution is $\pi_{\text{te}}$, we can regard that $N_{\text{te}}\pi_{\text{te}}$ positive data are included in the $N_{\text{te}}$ unlabeled data. Thus, when the $N_{\text{te}}$ unlabelled data are sorted by score, the top $N_{\text{te}}\pi_{\text{te}}$ instances can be considered positive (assuming this scoring is accurate), and the score of the $N_{\text{te}}\pi_{\text{te}}$-th instance can be the boundary separating positive and negative. Thus, we can use the score of the $N_{\text{te}}\pi_{\text{te}}$-th instance as the threshold. Since true prior $\pi_{\text{te}}$ is unknown, we want to estimate it and use the estimated prior $\pi_{\text{te}}^{\text{est}}$ instead.

Next, we explain the procedure to obtain the estimated class-prior in the test distribution $\pi_{\text{te}}^{\text{est}}$. Specifically, we first extract negative data in unlabeled data from the training distribution by applying (off-the-shelf) PU learning to PU data in the training distribution. Since PU data and the class-prior in the training distribution are available in our setting, we can perform it. Then, since the negative distribution does not change in our setting, the extracted negative data can be regarded as negative data in the test distribution. We can estimate the class-prior in the test distribution by applying existing class-prior estimation methods [44, 63, 15] to the extracted negative and unlabeled data in the test distribution.

## D   Discussion about Negative Distribution Shift

In the main paper, we considered the positive distribution shift. Here, we consider a negative distribution shift, i.e., $p_{\text{te}}^{\text{p}}(\mathbf{x}) = p_{\text{tr}}^{\text{p}}(\mathbf{x})$ but $p_{\text{te}}^{\text{n}}(\mathbf{x}) \neq p_{\text{tr}}^{\text{n}}(\mathbf{x})$. In this case, when positive data in the training distribution and unlabeled data in the test distribution are available, we show that the negative distribution shift can be addressed by using existing AUC maximization methods [45, 58, 59].

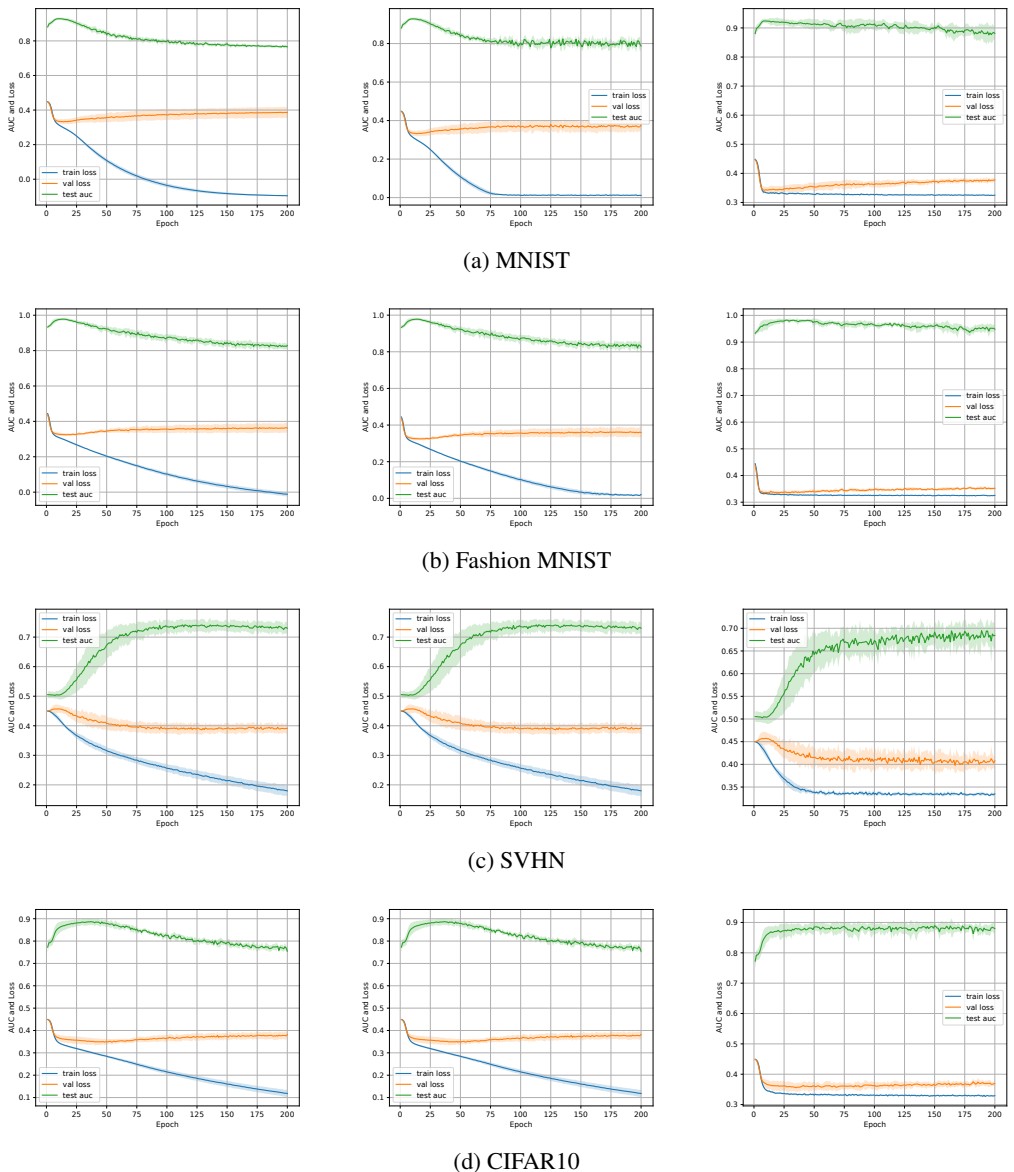

(a) MNIST

(b) Fashion MNIST

(c) SVHN

(d) CIFAR10

Figure 6: Dynamics of training loss, validation loss, and test AUC when $(\pi_{\mathrm{tr}}, \pi_{\mathrm{te}}) = (0.1, 0.3)$. Each column represents the results of Ours, Ours w/ nn, and Ours w/ b, respectively from left to right. The value of $b = (1 - \pi_{\mathrm{tr}})(1 - \pi_{\mathrm{te}})/2$ is 0.315 for all datasets.

Specifically, in this case, the AUC in the test distribution is

$$\mathrm{AUC}_\sigma(s) = \mathbb{E}_{\mathbf{x}^{\mathrm{P}} \sim p_{\mathrm{te}}^{\mathrm{P}}(\mathbf{x})} \mathbb{E}_{\mathbf{x}^{\mathrm{n}} \sim p_{\mathrm{te}}^{\mathrm{n}}(\mathbf{x})} \left[ f(\mathbf{x}^{\mathrm{P}}, \mathbf{x}^{\mathrm{n}}) \right] = \frac{1}{1 - \pi_{\mathrm{te}}} \mathbb{E}_{\mathbf{x}^{\mathrm{P}} \sim p_{\mathrm{tr}}^{\mathrm{P}}(\mathbf{x})} \mathbb{E}_{\mathbf{x} \sim p_{\mathrm{te}}(\mathbf{x})} \left[ f(\mathbf{x}^{\mathrm{P}}, \mathbf{x}^{\mathrm{n}}) \right] + C, \tag{21}$$

where $C$ is a constant, and in the second equal sign, we used that $p_{\mathrm{te}}^{\mathrm{n}}(\mathbf{x}) = \frac{1}{1 - \pi_{\mathrm{te}}} [p_{\mathrm{te}}(\mathbf{x}) - \pi_{\mathrm{te}} p_{\mathrm{te}}^{\mathrm{P}}(\mathbf{x})]$, $p_{\mathrm{te}}^{\mathrm{P}}(\mathbf{x}) = p_{\mathrm{tr}}^{\mathrm{P}}(\mathbf{x})$, and the fact that the AUC between the same densities is a constant as described in Section 4.2. This derived AUC is equivalent to that in the existing studies [45, 58, 59] and can be maximized with test unlabeled and training positive data. Note that we do not require to know $\pi_{\mathrm{te}}$ since it does not affect the optimization.

Table 6: Comparison with BPURR: average test AUCs for each test class-prior. Values in bold are not statistically different at the $5\%$ level from the best performing method in each row according to a paired t-test. FMNIST is an acronym for FashionMNIST.

| Data | MNIST | | | FMNIST | | | SVHN | | | CIFAR10 | | |
|------|-------|-------|-------|--------|-------|-------|-------|-------|-------|---------|-------|-------|
| $\pi_{\mathrm{te}}$ | 0.1 | 0.2 | 0.3 | 0.1 | 0.2 | 0.3 | 0.1 | 0.2 | 0.3 | 0.1 | 0.2 | 0.3 |
| Ours | 0.723 | **0.854** | **0.902** | 0.787 | **0.891** | **0.960** | **0.554** | **0.660** | **0.736** | **0.727** | **0.825** | **0.873** |
| BPURR | **0.863** | **0.904** | **0.907** | **0.939** | **0.929** | 0.917 | 0.495 | 0.516 | 0.532 | **0.745** | 0.786 | 0.790 |

## E  Additional Experimental Results

### E.1  Comparison with PURR using Class Balanced Classification Loss

In the comparison methods, PURR is the PU learning method under the positive distribution shift to minimize the test classification risk [16]. Since the classification risk is not suitable for imbalanced classification, we created a new method of PURR that uses the class balanced classification loss (the balanced class rate) [6] instead of the classification risk, called BPURR. Even in this case, BPURR requires the class-prior on the test distribution, which is usually difficult to know and is not required in the proposed method. Table 6 shows the mean test AUC with each class-prior on the test distribution. BPURR used the class-prior on the training distribution as that on the test distribution. When $\pi_{\mathrm{te}} = 0.1$, BPURR worked well since the used class-prior was equal to the true value. However, in other cases, BPURR performed worse than the proposed method in many cases.

### E.2  Results with Small Class-priors on the Training Distribution

In the main paper, the positive class-prior on the training distribution $\pi_{\mathrm{tr}}$ was set to $0.1$. Here, we investigated how the performance of the proposed method changes when $\pi_{\mathrm{tr}}$ is smaller. Table 7 shows the average test AUCs of each method on the four datasets. The proposed method performed the best or comparably to it in almost all cases (32 out of 36). As the positive class-prior on the training distribution $\pi_{\mathrm{tr}}$ decreased, the proposed method tended to perform well. Since the amount of noise (positive data in the training distribution) in $X_{\mathrm{tr}}$ decreases as $\pi_{\mathrm{tr}}$ decreases, negative data (which is essential information to maximize the AUC on the test distribution) might be easy to extract from $X_{\mathrm{tr}}$ when $\pi_{\mathrm{tr}}$ is small.

### E.3  Results with Tabular Datasets

We evaluate the proposed method with two tabular datasets with distribution shifts (HReadmission and Hypertension) [14]. In HReadmission, the task is to predict the 30-day readmission of diabetic hospital patients. Each patient is represented by a $183$-dimensional feature vector. In Hypertension, the task is a hypertension diagnosis for high-risk age. Each survey subject is represented by a $100$-dimensional feature vector. We constructed the positive distribution shift using positive-shifted data (positive ood data) and non-shifted data (positive and negative training data). The experimental setting, such as the number of data, is the same as that in the main paper. Table 8 shows the result. The proposed method outperformed the others. This result shows that the proposed method is effective for tabular datasets.

### E.4  Results with Standard Deviations

In the main paper, we omitted the standard deviations of results due to the limited space. Table 9 shows the mean test AUCs with the standard deviations.

### E.5  F1 Score and G-mean of TPR and TNR

Although this paper focuses on maximizing the AUC since it is a representative evaluation metric for imbalanced data, other metrics for imbalanced data exist, such as the F1 score and the G-mean of TPR and TNR [17]. Therefore, we investigated whether the proposed method is also effective for the F1 score and the G-mean of TPR and TNR. Unlike the AUC, the F1 score and the G-mean of TPR and TNR require to determine the classification threshold. In this experiment, when $N$ test data, in which the ratio of positive data is $\pi_{\mathrm{te}}$, are sorted by score of each method, we regard the top $N\pi_{\mathrm{te}}$

Table 7: Average test AUCs with different class-prior pairs. Values in bold are not statistically different at the $5\%$ level from the best performing method in each row according to a paired t-test. '# best' row represents the number of results of each method that are the best or comparable to it.

| Data | $\pi_{\mathrm{tr}}$ | $\pi_{\mathrm{te}}$ | Ours | CE | nnPU | puAUC | PULA | AnnPU | ApuAUC | APULA | CpuAUC | PURR |
|---|---|---|---|---|---|---|---|---|---|---|---|---|
| MNIST | 0.01 | 0.1 | **0.871** | 0.813 | 0.724 | 0.814 | 0.811 | 0.786 | 0.805 | 0.803 | 0.818 | **0.895** |
| | 0.01 | 0.2 | 0.923 | 0.813 | 0.724 | 0.814 | 0.811 | 0.786 | 0.798 | 0.783 | 0.817 | **0.948** |
| | 0.01 | 0.3 | **0.953** | 0.813 | 0.724 | 0.814 | 0.811 | 0.788 | 0.792 | 0.757 | 0.817 | **0.957** |
| | 0.05 | 0.1 | 0.741 | 0.806 | 0.768 | 0.807 | 0.808 | 0.782 | 0.802 | 0.800 | 0.811 | **0.871** |
| | 0.05 | 0.2 | 0.869 | 0.806 | 0.772 | 0.807 | 0.808 | 0.794 | 0.796 | 0.775 | 0.811 | **0.921** |
| | 0.05 | 0.3 | **0.936** | 0.806 | 0.767 | 0.807 | 0.808 | 0.789 | 0.790 | 0.755 | 0.811 | **0.943** |
| | 0.1 | 0.1 | **0.723** | **0.796** | **0.782** | **0.801** | **0.802** | **0.782** | **0.798** | **0.793** | **0.804** | 0.815 |
| | 0.1 | 0.2 | **0.854** | 0.796 | 0.784 | 0.801 | 0.802 | 0.785 | 0.793 | 0.769 | 0.803 | **0.871** |
| | 0.1 | 0.3 | **0.902** | 0.796 | 0.783 | 0.800 | 0.802 | 0.780 | 0.786 | 0.745 | 0.802 | **0.914** |
| Fashion MNIST | 0.01 | 0.1 | **0.969** | **0.955** | 0.776 | **0.949** | 0.917 | 0.723 | 0.901 | **0.947** | **0.951** | 0.934 |
| | 0.01 | 0.2 | **0.978** | 0.955 | 0.777 | 0.949 | 0.917 | 0.745 | 0.872 | 0.926 | 0.954 | 0.936 |
| | 0.01 | 0.3 | **0.985** | 0.955 | 0.777 | 0.949 | 0.916 | 0.785 | 0.845 | 0.903 | 0.955 | 0.955 |
| | 0.05 | 0.1 | **0.925** | 0.935 | 0.796 | 0.950 | **0.973** | 0.740 | 0.902 | **0.947** | 0.950 | 0.928 |
| | 0.05 | 0.2 | **0.968** | 0.942 | 0.797 | 0.948 | **0.973** | 0.715 | 0.872 | **0.923** | 0.948 | 0.944 |
| | 0.05 | 0.3 | **0.982** | 0.933 | 0.796 | 0.949 | **0.973** | 0.792 | 0.843 | 0.904 | 0.949 | **0.962** |
| | 0.1 | 0.1 | 0.787 | 0.932 | 0.825 | 0.937 | **0.954** | 0.772 | 0.920 | **0.939** | 0.943 | 0.920 |
| | 0.1 | 0.2 | **0.891** | 0.930 | 0.825 | 0.937 | **0.955** | 0.771 | 0.890 | **0.911** | 0.943 | **0.929** |
| | 0.1 | 0.3 | **0.960** | 0.930 | 0.825 | **0.938** | **0.955** | 0.866 | 0.870 | 0.863 | **0.944** | **0.961** |
| SVHN | 0.01 | 0.1 | **0.586** | 0.503 | 0.503 | **0.522** | 0.499 | 0.503 | **0.521** | 0.500 | **0.538** | 0.501 |
| | 0.01 | 0.2 | **0.752** | 0.503 | 0.503 | 0.521 | 0.499 | 0.503 | 0.520 | 0.500 | 0.533 | 0.501 |
| | 0.01 | 0.3 | **0.801** | 0.503 | 0.503 | 0.521 | 0.499 | 0.503 | 0.517 | 0.497 | 0.532 | 0.495 |
| | 0.05 | 0.1 | **0.587** | 0.503 | 0.501 | 0.513 | 0.498 | 0.501 | 0.514 | 0.498 | 0.510 | 0.501 |
| | 0.05 | 0.2 | **0.715** | 0.505 | 0.501 | 0.513 | 0.498 | 0.501 | 0.513 | 0.495 | 0.517 | 0.501 |
| | 0.05 | 0.3 | **0.779** | 0.505 | 0.501 | 0.513 | 0.498 | 0.501 | 0.512 | 0.493 | 0.517 | 0.501 |
| | 0.1 | 0.1 | **0.554** | 0.504 | 0.501 | 0.518 | 0.494 | 0.501 | 0.512 | 0.492 | 0.524 | 0.511 |
| | 0.1 | 0.2 | **0.660** | 0.503 | 0.501 | 0.518 | 0.494 | 0.501 | 0.507 | 0.490 | 0.523 | 0.522 |
| | 0.1 | 0.3 | **0.736** | 0.503 | 0.501 | 0.518 | 0.494 | 0.501 | 0.507 | 0.490 | 0.523 | 0.519 |
| CIFAR10 | 0.01 | 0.1 | **0.805** | 0.738 | 0.414 | 0.748 | **0.765** | 0.454 | 0.746 | 0.724 | 0.738 | 0.468 |
| | 0.01 | 0.2 | **0.883** | 0.743 | 0.414 | 0.749 | 0.765 | 0.534 | 0.719 | 0.748 | 0.757 | 0.593 |
| | 0.01 | 0.3 | **0.898** | 0.741 | 0.414 | 0.749 | 0.765 | 0.532 | 0.721 | 0.724 | 0.733 | **0.914** |
| | 0.05 | 0.1 | **0.740** | 0.736 | 0.419 | **0.745** | 0.724 | 0.478 | **0.737** | **0.729** | **0.751** | 0.489 |
| | 0.05 | 0.2 | **0.861** | 0.737 | 0.419 | 0.745 | 0.724 | 0.520 | 0.723 | 0.731 | 0.750 | 0.518 |
| | 0.05 | 0.3 | **0.894** | 0.740 | 0.419 | 0.745 | 0.724 | 0.536 | 0.712 | 0.736 | 0.750 | **0.845** |
| | 0.1 | 0.1 | **0.727** | **0.682** | 0.455 | **0.749** | **0.751** | 0.489 | **0.739** | **0.950** | **0.750** | 0.434 |
| | 0.1 | 0.2 | **0.825** | 0.679 | 0.467 | 0.741 | 0.739 | 0.491 | 0.732 | 0.744 | 0.744 | 0.536 |
| | 0.1 | 0.3 | **0.874** | 0.682 | 0.451 | 0.750 | 0.738 | 0.545 | 0.722 | 0.749 | 0.749 | 0.710 |
| # best | | | 32 | 4 | 1 | 4 | 10 | 1 | 4 | 8 | 5 | 14 |

Table 8: Results with tabular data: average test AUCs over different class-prior pairs $\pi_{\mathrm{te}}$ in the test distribution within $\{0.1, 0.2, 0.3\}$ when training class-prior $\pi_{\mathrm{tr}}$ is 0.1. Boldface denotes the best and comparable methods according to the paired t-test and the significance level of $5\%$.

| Data | Ours | CE | nnPU | puAUC | PULA | AnnPU | ApuAUC | APULA | CpuAUC | PURR |
|---|---|---|---|---|---|---|---|---|---|---|
| HReadmission | **0.747** | 0.529 | 0.562 | 0.502 | 0.519 | 0.531 | 0.492 | 0.495 | 0.511 | 0.695 |
| Hypertension | **0.630** | 0.596 | 0.607 | 0.571 | 0.571 | 0.565 | 0.608 | 0.539 | 0.571 | **0.643** |

data as positive and the remaining as negative. Tables 10 and 11 show the average test F1 scores and G-means of TPR and TNR with different test and training class-prior pairs, respectively. The proposed method tended to outperform the other methods although it maximizes the AUC. This result may imply that AUC maximization can also help to improve other evaluation metrics for imbalanced data.

Table 9: Average test AUCs and standard deviations with different class-prior pairs.

| Data | $\pi_{te}$ | Ours | CE | nnPU | puAUC | PULA |
|---|---|---|---|---|---|---|
| MNIST | 0.1 | 0.723(0.094) | 0.796(0.014) | 0.782(0.034) | 0.801(0.011) | 0.802(0.020) |
| | 0.2 | 0.854(0.062) | 0.796(0.014) | 0.784(0.035) | 0.801(0.010) | 0.802(0.020) |
| | 0.3 | 0.902(0.034) | 0.796(0.013) | 0.783(0.035) | 0.800(0.011) | 0.802(0.020) |
| Fashion | 0.1 | 0.787(0.091) | 0.932(0.020) | 0.825(0.075) | 0.937(0.038) | 0.954(0.030) |
| MNIST | 0.2 | 0.891(0.071) | 0.930(0.021) | 0.825(0.075) | 0.937(0.038) | 0.955(0.030) |
| | 0.3 | 0.960(0.037) | 0.930(0.020) | 0.825(0.075) | 0.938(0.038) | 0.955(0.030) |
| SVHN | 0.1 | 0.554(0.045) | 0.504(0.015) | 0.501(0.009) | 0.518(0.027) | 0.494(0.013) |
| | 0.2 | 0.660(0.047) | 0.503(0.016) | 0.501(0.009) | 0.518(0.027) | 0.494(0.013) |
| | 0.3 | 0.736(0.013) | 0.503(0.015) | 0.501(0.009) | 0.518(0.027) | 0.494(0.013) |
| CIFAR10 | 0.1 | 0.727(0.070) | 0.682(0.090) | 0.455(0.098) | 0.749(0.050) | 0.751(0.029) |
| | 0.2 | 0.825(0.048) | 0.679(0.100) | 0.467(0.121) | 0.741(0.055) | 0.739(0.047) |
| | 0.3 | 0.874(0.022) | 0.682(0.090) | 0.451(0.082) | 0.750(0.050) | 0.738(0.048) |

| Data | $\pi_{te}$ | APULA | AnnPU | ApuAUC | CpuAUC | PURR |
|---|---|---|---|---|---|---|
| MNIST | 0.1 | 0.793(0.021) | 0.782(0.035) | 0.798(0.009) | 0.804(0.011) | 0.815(0.037) |
| | 0.2 | 0.769(0.017) | 0.785(0.017) | 0.793(0.009) | 0.803(0.011) | 0.871(0.014) |
| | 0.3 | 0.745(0.017) | 0.780(0.017) | 0.786(0.010) | 0.802(0.010) | 0.914(0.014) |
| Fashion | 0.1 | 0.939(0.066) | 0.772(0.073) | 0.920(0.046) | 0.943(0.025) | 0.920(0.010) |
| MNIST | 0.2 | 0.911(0.082) | 0.771(0.104) | 0.890(0.059) | 0.943(0.026) | 0.929(0.011) |
| | 0.3 | 0.863(0.098) | 0.866(0.105) | 0.870(0.049) | 0.944(0.026) | 0.961(0.025) |
| SVHN | 0.1 | 0.492(0.013) | 0.501(0.009) | 0.512(0.022) | 0.524(0.026) | 0.511(0.009) |
| | 0.2 | 0.490(0.012) | 0.501(0.009) | 0.507(0.015) | 0.523(0.026) | 0.522(0.009) |
| | 0.3 | 0.490(0.008) | 0.501(0.009) | 0.507(0.017) | 0.523(0.025) | 0.519(0.008) |
| CIFAR10 | 0.1 | 0.750(0.029) | 0.489(0.141) | 0.739(0.055) | 0.750(0.051) | 0.434(0.085) |
| | 0.2 | 0.742(0.038) | 0.491(0.139) | 0.732(0.056) | 0.744(0.051) | 0.536(0.162) |
| | 0.3 | 0.750(0.044) | 0.545(0.144) | 0.722(0.060) | 0.749(0.049) | 0.710(0.197) |

Table 10: Average test F1 scores with different class-prior pairs. We set training class-prior $\pi_{tr}$ to $0.1$. Values in bold are not statistically different at the $5\%$ level from the best performing method in each row according to a paired t-test. '# best' row represents the number of results of each method that are the best or comparable to it.

| Data | $\pi_{te}$ | Ours | CE | nnPU | puAUC | PULA | AnnPU | ApuAUC | APULA | CpuAUC | PURR |
|---|---|---|---|---|---|---|---|---|---|---|---|
| MNIST | 0.1 | **0.395** | **0.445** | **0.432** | **0.443** | **0.425** | **0.418** | **0.437** | 0.383 | **0.445** | **0.469** |
| | 0.2 | **0.637** | 0.548 | 0.534 | 0.551 | 0.541 | 0.504 | 0.538 | 0.454 | 0.553 | **0.645** |
| | 0.3 | **0.750** | 0.618 | 0.603 | 0.621 | 0.621 | 0.581 | 0.608 | 0.525 | 0.629 | **0.782** |
| Fashion | 0.1 | 0.619 | **0.702** | 0.445 | **0.727** | **0.778** | 0.342 | 0.683 | **0.738** | **0.746** | **0.746** |
| MNIST | 0.2 | **0.763** | 0.755 | 0.545 | 0.786 | **0.844** | 0.667 | 0.687 | 0.722 | 0.794 | **0.792** |
| | 0.3 | **0.883** | 0.793 | 0.631 | **0.821** | **0.874** | 0.718 | 0.687 | 0.697 | **0.830** | **0.883** |
| SVHN | 0.1 | **0.163** | 0.110 | 0.111 | 0.112 | 0.101 | 0.111 | 0.121 | 0.097 | 0.129 | 0.116 |
| | 0.2 | **0.368** | 0.207 | 0.201 | 0.217 | 0.195 | 0.206 | 0.211 | 0.193 | 0.227 | 0.208 |
| | 0.3 | **0.542** | 0.310 | 0.310 | 0.323 | 0.293 | 0.309 | 0.314 | 0.287 | 0.324 | 0.321 |
| CIFAR10 | 0.1 | **0.337** | **0.299** | 0.090 | **0.381** | **0.371** | 0.120 | **0.367** | **0.370** | **0.381** | 0.070 |
| | 0.2 | **0.574** | 0.401 | 0.181 | 0.493 | 0.491 | 0.201 | 0.482 | 0.484 | 0.498 | 0.260 |
| | 0.3 | **0.716** | 0.489 | 0.235 | 0.583 | 0.570 | 0.348 | 0.551 | 0.566 | 0.583 | 0.531 |
| # best | | 11 | 3 | 1 | 4 | 5 | 1 | 2 | 2 | 4 | 6 |


Table 11: Average test G-mean of TPR and TNR with different class-prior pairs. We set training class-prior $\pi_{\mathrm{tr}}$ to 0.1. Values in bold are not statistically different at the 5% level from the best performing method in each row according to a paired t-test. '# best' row represents the number of results of each method that are the best or comparable to it.

| Data | $\pi_{\mathrm{te}}$ | Ours | CE | nnPU | puAUC | PULA | AnnPU | ApuAUC | APULA | CpuAUC | PURR |
|---|---|---|---|---|---|---|---|---|---|---|---|
| MNIST | 0.1 | **0.604** | **0.646** | **0.634** | **0.644** | **0.629** | **0.624** | **0.640** | 0.596 | **0.646** | **0.663** |
| | 0.2 | **0.760** | 0.697 | 0.683 | 0.699 | 0.692 | 0.664 | 0.690 | 0.626 | 0.700 | **0.767** |
| | 0.3 | **0.818** | 0.719 | 0.707 | 0.721 | 0.721 | 0.690 | 0.711 | 0.647 | 0.721 | **0.842** |
| FMNIST | 0.1 | 0.767 | **0.823** | 0.636 | **0.837** | **0.868** | 0.555 | 0.809 | **0.832** | **0.850** | 0.746 |
| | 0.2 | **0.846** | 0.842 | 0.690 | 0.861 | **0.899** | 0.628 | 0.780 | 0.812 | 0.866 | **0.866** |
| | 0.3 | **0.915** | 0.850 | 0.727 | **0.870** | **0.909** | 0.792 | 0.771 | 0.776 | **0.877** | **0.916** |
| SVHN | 0.1 | **0.382** | 0.313 | 0.315 | 0.328 | 0.300 | 0.315 | 0.330 | 0.294 | 0.340 | 0.322 |
| | 0.2 | **0.555** | 0.407 | 0.406 | 0.417 | 0.394 | 0.406 | 0.411 | 0.392 | 0.427 | 0.408 |
| | 0.3 | **0.660** | 0.468 | 0.467 | 0.479 | 0.451 | 0.466 | 0.470 | 0.446 | 0.480 | 0.477 |
| CIFAR10 | 0.1 | **0.556** | **0.519** | 0.258 | **0.595** | **0.587** | 0.292 | **0.582** | **0.586** | **0.595** | 0.229 |
| | 0.2 | **0.716** | 0.578 | 0.363 | 0.655 | 0.654 | 0.388 | 0.646 | 0.648 | 0.659 | 0.169 |
| | 0.3 | **0.793** | 0.616 | 0.394 | 0.691 | 0.682 | 0.493 | 0.667 | 0.678 | 0.691 | 0.643 |
| # best | | 11 | 3 | 1 | 4 | 5 | 1 | 2 | 2 | 4 | 6 |

- The abstract and/or introduction should clearly state the claims made, including the contributions made in the paper and important assumptions and limitations. A No or NA answer to this question will not be perceived well by the reviewers.
- The claims made should match theoretical and experimental results, and reflect how much the results can be expected to generalize to other settings.
- It is fine to include aspirational goals as motivation as long as it is clear that these goals are not attained by the paper.

2. **Limitations**

   Question: Does the paper discuss the limitations of the work performed by the authors?

   Answer: [Yes]

   Justification: The limitations of our work are discussed in Section 7.

   Guidelines:

   - The answer NA means that the paper has no limitation while the answer No means that the paper has limitations, but those are not discussed in the paper.
   - The authors are encouraged to create a separate "Limitations" section in their paper.
   - The paper should point out any strong assumptions and how robust the results are to violations of these assumptions (e.g., independence assumptions, noiseless settings, model well-specification, asymptotic approximations only holding locally). The authors should reflect on how these assumptions might be violated in practice and what the implications would be.
   - The authors should reflect on the scope of the claims made, e.g., if the approach was only tested on a few datasets or with a few runs. In general, empirical results often depend on implicit assumptions, which should be articulated.
   - The authors should reflect on the factors that influence the performance of the approach. For example, a facial recognition algorithm may perform poorly when image resolution is low or images are taken in low lighting. Or a speech-to-text system might not be used reliably to provide closed captions for online lectures because it fails to handle technical jargon.
   - The authors should discuss the computational efficiency of the proposed algorithms and how they scale with dataset size.
   - If applicable, the authors should discuss possible limitations of their approach to address problems of privacy and fairness.
   - While the authors might fear that complete honesty about limitations might be used by reviewers as grounds for rejection, a worse outcome might be that reviewers discover limitations that aren't acknowledged in the paper. The authors should use their best judgment and recognize that individual actions in favor of transparency play an important role in developing norms that preserve the integrity of the community. Reviewers will be specifically instructed to not penalize honesty concerning limitations.

3. **Theory Assumptions and Proofs**

   Question: For each theoretical result, does the paper provide the full set of assumptions and a complete (and correct) proof?

   Answer: [Yes]

   Justification: The full sets of assumptions and proof are described in Section 4.

   Guidelines:

   - The answer NA means that the paper does not include theoretical results.
   - All the theorems, formulas, and proofs in the paper should be numbered and cross-referenced.
   - All assumptions should be clearly stated or referenced in the statement of any theorems.
   - The proofs can either appear in the main paper or the supplemental material, but if they appear in the supplemental material, the authors are encouraged to provide a short proof sketch to provide intuition.
   - Inversely, any informal proof provided in the core of the paper should be complemented by formal proofs provided in appendix or supplemental material.
   - Theorems and Lemmas that the proof relies upon should be properly referenced.

4. **Experimental Result Reproducibility**

   Question: Does the paper fully disclose all the information needed to reproduce the main experimental results of the paper to the extent that it affects the main claims and/or conclusions of the paper (regardless of whether the code and data are provided or not)?

   Answer: [Yes]

   Justification: The details of the experimental settings (data, comparison methods, network architecture, and hyperparameters) are described in Section 5.

   Guidelines:

   - The answer NA means that the paper does not include experiments.
   - If the paper includes experiments, a No answer to this question will not be perceived well by the reviewers: Making the paper reproducible is important, regardless of whether the code and data are provided or not.
   - If the contribution is a dataset and/or model, the authors should describe the steps taken to make their results reproducible or verifiable.
   - Depending on the contribution, reproducibility can be accomplished in various ways. For example, if the contribution is a novel architecture, describing the architecture fully might suffice, or if the contribution is a specific model and empirical evaluation, it may be necessary to either make it possible for others to replicate the model with the same dataset, or provide access to the model. In general. releasing code and data is often one good way to accomplish this, but reproducibility can also be provided via detailed instructions for how to replicate the results, access to a hosted model (e.g., in the case of a large language model), releasing of a model checkpoint, or other means that are appropriate to the research performed.
   - While NeurIPS does not require releasing code, the conference does require all submissions to provide some reasonable avenue for reproducibility, which may depend on the nature of the contribution. For example
     (a) If the contribution is primarily a new algorithm, the paper should make it clear how to reproduce that algorithm.
     (b) If the contribution is primarily a new model architecture, the paper should describe the architecture clearly and fully.
     (c) If the contribution is a new model (e.g., a large language model), then there should either be a way to access this model for reproducing the results or a way to reproduce the model (e.g., with an open-source dataset or instructions for how to construct the dataset).
     (d) We recognize that reproducibility may be tricky in some cases, in which case authors are welcome to describe the particular way they provide for reproducibility. In the case of closed-source models, it may be that access to the model is limited in some way (e.g., to registered users), but it should be possible for other researchers to have some path to reproducing or verifying the results.

5. **Open access to data and code**

   Question: Does the paper provide open access to the data and code, with sufficient instructions to faithfully reproduce the main experimental results, as described in supplemental material?

   Answer: [No]

   Justification: The code is proprietary. The datasets and experimental procedures are described in Section 5. We described the pseudocode of our method in Section 4.

   Guidelines:

   - The answer NA means that paper does not include experiments requiring code.
   - Please see the NeurIPS code and data submission guidelines (`https://nips.cc/public/guides/CodeSubmissionPolicy`) for more details.
   - While we encourage the release of code and data, we understand that this might not be possible, so "No" is an acceptable answer. Papers cannot be rejected simply for not including code, unless this is central to the contribution (e.g., for a new open-source benchmark).
   - The instructions should contain the exact command and environment needed to run to reproduce the results. See the NeurIPS code and data submission guidelines (`https://nips.cc/public/guides/CodeSubmissionPolicy`) for more details.
   - The authors should provide instructions on data access and preparation, including how to access the raw data, preprocessed data, intermediate data, and generated data, etc.
   - The authors should provide scripts to reproduce all experimental results for the new proposed method and baselines. If only a subset of experiments are reproducible, they should state which ones are omitted from the script and why.
   - At submission time, to preserve anonymity, the authors should release anonymized versions (if applicable).
   - Providing as much information as possible in supplemental material (appended to the paper) is recommended, but including URLs to data and code is permitted.

6. **Experimental Setting/Details**

   Question: Does the paper specify all the training and test details (e.g., data splits, hyperparameters, how they were chosen, type of optimizer, etc.) necessary to understand the results?

   Answer: [Yes]

   Justification: The details of the experimental settings are described in Section 5.

   Guidelines:

   - The answer NA means that the paper does not include experiments.
   - The experimental setting should be presented in the core of the paper to a level of detail that is necessary to appreciate the results and make sense of them.
   - The full details can be provided either with the code, in appendix, or as supplemental material.

7. **Experiment Statistical Significance**

   Question: Does the paper report error bars suitably and correctly defined or other appropriate information about the statistical significance of the experiments?

   Answer: [Yes]

   Justification: We conducted a statistical significance test (paired t-test) in tables 1, 6, and 7. The standard deviations of Table 1 are reported in Table 9. The standard errors are reported in all graphs.

   Guidelines:

   - The answer NA means that the paper does not include experiments.
   - The authors should answer "Yes" if the results are accompanied by error bars, confidence intervals, or statistical significance tests, at least for the experiments that support the main claims of the paper.

- The factors of variability that the error bars are capturing should be clearly stated (for example, train/test split, initialization, random drawing of some parameter, or overall run with given experimental conditions).
- The method for calculating the error bars should be explained (closed form formula, call to a library function, bootstrap, etc.)
- The assumptions made should be given (e.g., Normally distributed errors).
- It should be clear whether the error bar is the standard deviation or the standard error of the mean.
- It is OK to report 1-sigma error bars, but one should state it. The authors should preferably report a 2-sigma error bar than state that they have a 96% CI, if the hypothesis of Normality of errors is not verified.
- For asymmetric distributions, the authors should be careful not to show in tables or figures symmetric error bars that would yield results that are out of range (e.g. negative error rates).
- If error bars are reported in tables or plots, The authors should explain in the text how they were calculated and reference the corresponding figures or tables in the text.

8. **Experiments Compute Resources**

Question: For each experiment, does the paper provide sufficient information on the computer resources (type of compute workers, memory, time of execution) needed to reproduce the experiments?

Answer: [Yes]

Justification: We described the computer resource in Section 5.

Guidelines:

- The answer NA means that the paper does not include experiments.
- The paper should indicate the type of compute workers CPU or GPU, internal cluster, or cloud provider, including relevant memory and storage.
- The paper should provide the amount of compute required for each of the individual experimental runs as well as estimate the total compute.
- The paper should disclose whether the full research project required more compute than the experiments reported in the paper (e.g., preliminary or failed experiments that didn't make it into the paper).

9. **Code Of Ethics**

Question: Does the research conducted in the paper conform, in every respect, with the NeurIPS Code of Ethics https://neurips.cc/public/EthicsGuidelines?

Answer: [Yes]

Justification: We confirmed the NeurIPS Code of Ethnics.

Guidelines:

- The answer NA means that the authors have not reviewed the NeurIPS Code of Ethics.
- If the authors answer No, they should explain the special circumstances that require a deviation from the Code of Ethics.
- The authors should make sure to preserve anonymity (e.g., if there is a special consideration due to laws or regulations in their jurisdiction).

10. **Broader Impacts**

Question: Does the paper discuss both potential positive societal impacts and negative societal impacts of the work performed?

Answer: [Yes]

Justification: We discussed the potential social impacts in Section A.

Guidelines:

- The answer NA means that there is no societal impact of the work performed.
- If the authors answer NA or No, they should explain why their work has no societal impact or why the paper does not address societal impact.

- Examples of negative societal impacts include potential malicious or unintended uses (e.g., disinformation, generating fake profiles, surveillance), fairness considerations (e.g., deployment of technologies that could make decisions that unfairly impact specific groups), privacy considerations, and security considerations.
- The conference expects that many papers will be foundational research and not tied to particular applications, let alone deployments. However, if there is a direct path to any negative applications, the authors should point it out. For example, it is legitimate to point out that an improvement in the quality of generative models could be used to generate deepfakes for disinformation. On the other hand, it is not needed to point out that a generic algorithm for optimizing neural networks could enable people to train models that generate Deepfakes faster.
- The authors should consider possible harms that could arise when the technology is being used as intended and functioning correctly, harms that could arise when the technology is being used as intended but gives incorrect results, and harms following from (intentional or unintentional) misuse of the technology.
- If there are negative societal impacts, the authors could also discuss possible mitigation strategies (e.g., gated release of models, providing defenses in addition to attacks, mechanisms for monitoring misuse, mechanisms to monitor how a system learns from feedback over time, improving the efficiency and accessibility of ML).

11. **Safeguards**

Question: Does the paper describe safeguards that have been put in place for responsible release of data or models that have a high risk for misuse (e.g., pretrained language models, image generators, or scraped datasets)?

Answer: [NA]

Justification: This paper does not release such high-risk models or datasets.

Guidelines:

- The answer NA means that the paper poses no such risks.
- Released models that have a high risk for misuse or dual-use should be released with necessary safeguards to allow for controlled use of the model, for example by requiring that users adhere to usage guidelines or restrictions to access the model or implementing safety filters.
- Datasets that have been scraped from the Internet could pose safety risks. The authors should describe how they avoided releasing unsafe images.
- We recognize that providing effective safeguards is challenging, and many papers do not require this, but we encourage authors to take this into account and make a best faith effort.

12. **Licenses for existing assets**

Question: Are the creators or original owners of assets (e.g., code, data, models), used in the paper, properly credited and are the license and terms of use explicitly mentioned and properly respected?

Answer: [Yes]

Justification: Datasets used in this paper are cited in Section 5.

Guidelines:

- The answer NA means that the paper does not use existing assets.
- The authors should cite the original paper that produced the code package or dataset.
- The authors should state which version of the asset is used and, if possible, include a URL.
- The name of the license (e.g., CC-BY 4.0) should be included for each asset.
- For scraped data from a particular source (e.g., website), the copyright and terms of service of that source should be provided.
- If assets are released, the license, copyright information, and terms of use in the package should be provided. For popular datasets, `paperswithcode.com/datasets` has curated licenses for some datasets. Their licensing guide can help determine the license of a dataset.

- For existing datasets that are re-packaged, both the original license and the license of the derived asset (if it has changed) should be provided.
- If this information is not available online, the authors are encouraged to reach out to the asset's creators.

13. **New Assets**

Question: Are new assets introduced in the paper well documented and is the documentation provided alongside the assets?

Answer: [NA]

Justification: This paper does not provide new assets.

Guidelines:

- The answer NA means that the paper does not release new assets.
- Researchers should communicate the details of the dataset/code/model as part of their submissions via structured templates. This includes details about training, license, limitations, etc.
- The paper should discuss whether and how consent was obtained from people whose asset is used.
- At submission time, remember to anonymize your assets (if applicable). You can either create an anonymized URL or include an anonymized zip file.

14. **Crowdsourcing and Research with Human Subjects**

Question: For crowdsourcing experiments and research with human subjects, does the paper include the full text of instructions given to participants and screenshots, if applicable, as well as details about compensation (if any)?

Answer: [NA]

Justification: This paper does not involve crowdsourcing nor research with human subjects.

Guidelines:

- The answer NA means that the paper does not involve crowdsourcing nor research with human subjects.
- Including this information in the supplemental material is fine, but if the main contribution of the paper involves human subjects, then as much detail as possible should be included in the main paper.
- According to the NeurIPS Code of Ethics, workers involved in data collection, curation, or other labor should be paid at least the minimum wage in the country of the data collector.

15. **Institutional Review Board (IRB) Approvals or Equivalent for Research with Human Subjects**

Question: Does the paper describe potential risks incurred by study participants, whether such risks were disclosed to the subjects, and whether Institutional Review Board (IRB) approvals (or an equivalent approval/review based on the requirements of your country or institution) were obtained?

Answer: [NA]

Justification: This paper does not involve crowdsourcing nor research with human subjects.

Guidelines:

- The answer NA means that the paper does not involve crowdsourcing nor research with human subjects.
- Depending on the country in which research is conducted, IRB approval (or equivalent) may be required for any human subjects research. If you obtained IRB approval, you should clearly state this in the paper.
- We recognize that the procedures for this may vary significantly between institutions and locations, and we expect authors to adhere to the NeurIPS Code of Ethics and the guidelines for their institution.
- For initial submissions, do not include any information that would break anonymity (if applicable), such as the institution conducting the review.

