# OpenReview forum: "AUC Maximization under Positive Distribution Shift"
_NeurIPS.cc/2024/Conference — NeurIPS 2024 poster_

### Official Review · Reviewer_CKbq · 2024-06-15

**Soundness:** 4
**Presentation:** 4
**Contribution:** 3
**Rating:** 6
**Confidence:** 5

**Summary:**

Due to a positive distribution shift, training and test distributions are not identical. However, existing AUC maximization methods don’t take it into account. To address this shift, this paper theoretically shows a new way to maximize the AUC on the test distribution by using positive and unlabeled data in the training distribution and unlabeled data in the test distribution. Finally, four real-world datasets validate the effectiveness of the proposed method.

**Strengths:**

-	The proposed setting is novel and practical in AUC optimization. The distribution of negative data is generally stable but the distribution of positive data is more diverse or time-varying in medical diagnosis, intrusion detection, and visual inspection.
-	The method presentation is easy to understand. This paper first introduces basic AUC fundamental knowledge. Then, it gives the problem setting of the proposed positive distribution shift. Based on this setting, the final expression is obtained through some intuitive and simple derivation. To be specific, the AUC maximization on the test distribution can be accomplished by using positive and unlabeled data in the training distribution and unlabeled data in the test distribution.

**Weaknesses:**

- The effect of the proposed methods on MINST and Fashion MINST datasets is not significant, which is inconsistent with those on the other datasets. The authors don’t give any explanation.
- The authors do not fully compare their method with the latest ones. For example,
  - Positive-Unlabeled Learning with Label Distribution Alignment. (TPAMI 2023)
  - Dist-PU: Positive-Unlabeled Learning from a Label Distribution Perspective. (CVPR 2022)
  - Positive-unlabeled learning using random forests via recursive greedy risk minimization. (NeurIPS 2022)
- All theoretical derivations are only based on the sigmoid surrogate loss. As far as I know, square loss is also popular. Can the theoretical results extend to the other losses?
- There are some typos. For example,
  - In line 105, “However, these all methods assume that” should be “However, all these methods assume that”.

**Questions:**

please refer to Weakness

**Limitations:**

Yes

---

> ### Author Rebuttal · Authors · 2024-08-06
>
> Thank you for your positive comments and constructive feedback.
>
> > The effect of the proposed methods on MINST and Fashion MINST datasets is not significant, which is inconsistent with those on the other datasets. The authors don’t give any explanation.
>
> As described in Line 279, since MNIST and Fashion MNIST are relatively simple data, the performances of our method and PURR that does not consider class-imbalance might be similar. More specifically, if the supports of positive and negative conditional densities are separated (i.e., data are simple), both losses of positive and negative data can be minimized without conflict between them, even when there is a class imbalance. Thus, class imbalance might not be a severe problem in this case. In contrast, when positive and negative-conditional densities overlap (i.e., complex data), minimizing losses for positive data located in the overlapped region causes to increase losses for negative data in the same region. Thus, when there is a class imbalance, positive data that are few can often be ignored. Therefore, in this case, class imbalance becomes a more severe problem. Since SVHN and CIFAR10 are more complex than MNIST and Fashion MNIST, our method worked well. We will clarify this.
>
> > The authors do not fully compare their method with the latest ones.
>
> Thank you for sharing relevant works.
> We additionally compared our method with the method [a] in your comment (PURDA).
> The results are described in Table 5 of the PDF file in the global response.
> PURDA used positive and unlabeled data in the training distribution in this experiment since it is designed for ordinary PU learning.
> Margin $\rho$ is selected from $\\{0.1,1,10\\}$ by validation data.
> Our method performed better than PURDA.
> This is because PURDA does not consider the distribution shift.
> All the methods in your comment do not consider the distribution shift and, thus, are unsuitable for our setting.
> In the final version, we will include this result and add all related works in your comments in Section 2.
>
> [a] Positive-Unlabeled Learning with Label Distribution Alignment. (TPAMI 2023)
>
> > All theoretical derivations are only based on the sigmoid surrogate loss. As far as I know, square loss is also popular. Can the theoretical results extend to the other losses?
>
> This is an insightful question. As long as we use symmetric losses (i.e., loss $l$ satisfying $l(z)+l(-z)=K$ for any $z \in \mathbb{R}$ and $K$ is a constant) [6], we can derive the same final loss function in Eq. 13. This is because the second term in Eq. 10 becomes constant, and thus, we can ignore it.
> We note that symmetric losses include a wide range of losses such as sigmoid, ramp, unhinged, and zero-one losses, although the squared loss is not included [6].
> We will include this discussion in the final version.
>
> > There are some typos.
>
> Thank you for pointing out the typos. We will again carefully proofread and prepare the final version.

---

> > ### Comment · Reviewer_CKbq · 2024-08-13
> >
> > Thanks for the rebuttal! The authors have clarified all of my concerns. Hence, I decide to keep my rating and increase my confidence.

---

> > > ### Author Response · Authors · 2024-08-13
> > >
> > > Thank you very much for your response. We are pleased to hear that you have increased the confidence level to 5. We sincerely appreciate the time and effort you have dedicated to reviewing and providing feedback on our paper.

---

### Official Review · Reviewer_dvdg · 2024-07-05

**Soundness:** 3
**Presentation:** 4
**Contribution:** 3
**Rating:** 6
**Confidence:** 3

**Summary:**

The paper proposes a method for AUC maximization in binary classification problems under positive distribution shift. They introduce their method, which is simple and easy to implement/understand, and then show it works well in some experiments.

**Strengths:**

- The paper is well written and easy to understand;
- The paper proposes a well-motivated method and show how it can be easily implemented in practice;
- The experiments are convincing.

**Weaknesses:**

- The authors do not discuss how the classification threshold can be chosen in a practical situation under positive distribution shift.

**Questions:**

How should the practitioner choose classification threshold after training their classifiers using your method?

**Limitations:**

The authors discuss limitations.

---

> ### Author Rebuttal · Authors · 2024-08-06
>
> Thank you for your positive comments and constructive feedback.
>
> > How should the practitioner choose classification threshold after training their classifiers using your method?
>
> Thank you for the insightful question.
> In practical use, there are many situations where it is beneficial just to be able to sort data in score order. For example, in anomaly detection, experts or operators can check data with high scores within the cost they can spend or until anomalous data do not appear. In disease diagnosis, patients with higher scores can be given priority for detailed examination.
> In recommendation systems, products can be presented to users in order of score.
>
> However, when a classification threshold is required, we can use the estimated class-prior in the test distribution to determine the threshold.
> Specifically, we first extract negative data in unlabeled data from the training distribution by applying (off-the-shelf) PU learning to PU data in the training distribution. Since PU data and the class-prior in the training distribution are available in our setting, we can perform it. Next, since the negative distribution does not change in our setting, the extracted negative data can be regarded as negative data in the test distribution. We can estimate the class-prior in the test distribution by applying existing class-prior estimation methods [40, 59, 14] to the extracted negative and unlabeled data in the test distribution. The threshold can be set to the top $N_{{\rm te}} \pi_{{\rm te}}^{{\rm est}}$-th score of unlabeled data, where $\pi_{{\rm te}}^{{\rm est}}$ is the estimated class-prior. We will include this discussion.

---

> > ### Comment · Reviewer_dvdg · 2024-08-09
> >
> > Thank you for your reply!
> >
> > 1. I agree with you sorting the data; that's a good point;
> > 2. I am a bit confused with your suggestion here. What is the purpose/rationale of setting the threshold to the $N_{te}\pi_{te}$-th score?

---

> > > ### Author Response · Authors · 2024-08-09
> > > **Thank you for your prompt response.**
> > >
> > > Thank you for your prompt response!
> > > We are pleased to hear that you agree with sorting the data. We will address this point in the final version.
> > >
> > > Here, we will explain the threshold in more detail.
> > >
> > > Now, we have $N_{{\rm te}}$ unlabeled data in the test distribution. When the true positive class prior (the ratio of positive data in unlabeled data) in the test distribution is $\pi_{{\rm te}}$, we can regard that $N_{{\rm te}} \pi_{{\rm te}}$ positive data are included in the $N_{{\rm te}}$ unlabeled data.
> > > Thus, when the $N_{{\rm te}}$ unlabelled data are sorted by score, the top $N_{{\rm te}} \pi_{{\rm te}}$ instances can be considered positive (assuming this scoring is accurate), and the score of the $N_{{\rm te}} \pi_{{\rm te}}$-th instance can be the boundary separating positive and negative. Thus, we can use the score of the $N_{{\rm te}} \pi_{{\rm te}}$-th instance as the threshold. Since true prior $\pi_{{\rm te}}$ is unknown, we used the estimated prior $\pi _{{\rm te}}^{{\rm est}}$ instead.
> > >
> > > Does this response meet your satisfaction?
> > > If you have any further questions or suggestions, please do not hesitate to contact us.

---

> > > > ### Comment · Reviewer_dvdg · 2024-08-09
> > > >
> > > > I think the main issue here is the following. When deciding on a classification threshold, we try to balance some metrics (e.g., recall and precision), by optimizing some combination of them. It is not clear why setting the threshold in the way you propose will solve the problem for all kinds of scenarios (sometimes you value more precision and sometimes you value more recall, for example). The problem is that I am not sure you can do that under the things you know or can estimate. If this is possible, that'd be a good direction to explore in a future version, if not, I would point it as a limitation. Does that make sense?
> > > >
> > > > It would be good to put more emphasis on the ordering aspect as well. In some circumstances, ordering is enough.

---

> > > > > ### Author Response · Authors · 2024-08-10
> > > > >
> > > > > Thank you for your valuable comment. We understood the intent of your question.
> > > > >
> > > > > If the estimated score function perfectly ranks the positives and negatives in the unlabeled data and the estimated number of positive data in the unlabeled data with the estimated class-prior is correct, then using the threshold determined by our method leads to recall=100% and precision=100%. Thus, this threshold would also be best for other metrics defined by the combination of recall and precision, such as the F1 score. Our threshold determination method is based on these principles.
> > > > >
> > > > > Of course, this is an ideal situation. In reality, an estimation error is involved.
> > > > > A small error may approximate recall=100% and precision=100%, but whether it works well in practice depends on the data and the application.
> > > > > Thus, if we want to more rigorously determine the threshold for each application (recall-oriented, precision-oriented, etc.), we would need extra knowledge, such as some labeled data in the test distribution. This could be a limitation of our method, but as there are no labels in the test distribution, it is not easy to make a strict adjustment.
> > > > > (Note that this is not specific to our method but is expected in methods where only unlabelled data is available in the test distribution (e.g., unsupervised domain adaptation and unsupervised anomaly detection)).
> > > > >
> > > > > Of course, this limitation is a case where a strict threshold is required. As you have acknowledged, there are many cases where just sorting data is sufficient.
> > > > >
> > > > > In the final version, we will incorporate your suggestion by emphasizing that our method can sort data and has many real-world applications. Additionally, we will mention that there is a way to determine the threshold by using estimating class prior. Then, we will describe that if a strict threshold is required for the application, additional information, such as some labels, is desirable in the limitation section. Finally, we will describe that research on methods that maximize desired metrics, such as the combination of recall and precision, other than AUC under our problem setting is a future challenge.

---

> > > > > > ### Comment · Reviewer_dvdg · 2024-08-12
> > > > > >
> > > > > > Thank you for your response. I think your paper is good, but discussing this issue we have been talking about will be an important part of your work.
> > > > > >
> > > > > > I will keep my score as "weak accept".

---

> > > > > > > ### Author Response · Authors · 2024-08-13
> > > > > > >
> > > > > > > Thank you for your reply.
> > > > > > >
> > > > > > > We are pleased that you recognize the quality of the paper. We agree that the practical approach to determining the thresholds, as discussed, is an important point.
> > > > > > >
> > > > > > > If you have any additional questions, please feel free to contact us.
> > > > > > > Thank you again for taking the time to peer review and discuss the paper.

---

### Official Review · Reviewer_4KuN · 2024-07-11

**Soundness:** 2
**Presentation:** 2
**Contribution:** 2
**Rating:** 5
**Confidence:** 4

**Summary:**

This paper considers AUC maximization when the conditional probability distribution of the positive class changes in the test phase. To this end, the unbiased loss function is derived. The loss is approximated by positive and unlabeled data from training distribution, unlabeled data from test distribution, and class-prior of training distribution. In experiments, the proposed method outperformed the existing methods over the four benchmark datasets.

**Strengths:**

- This is the first study on AUC maximization for positive distribution shift.
- The proposed method outperformed the existing methods.
- The proposed method does not require the class-prior of the test distribution.

**Weaknesses:**

- Unlike the existing study [15, 42], the negative distribution is not considered.
- It lacks theoretical analyses of the proposed method.
- The extension of the proposed method is discussed but not evaluated in the experiments.

**Questions:**

Is the proposed loss function unbiased to its supervised counterpart?

It would be valuable if there were discussion or experimental results showing the effect of the number of unlabeled data from the test distribution. In some applications, collecting a lot of unlabeled data from the test distribution might be difficult. In such a situation, the experimental results would help practitioners understand how many samples are necessary to collect.

According to the literature, the non-negative risk estimator plays a crucial role in training deep neural networks. However, the proposed method does not mention the non-negativity of the risk estimator. Did the authors encounter that the risk estimator went to a large negative value in experiments? If not, what points in the proposed method avoid the issue?

Regarding the Extension in Section 4.4, it would be nice to cite the existing work.

**Limitations:**

The limitations are discussed in Appendix A.

---

> ### Author Rebuttal · Authors · 2024-08-06
>
> Thank you for your insightful comments and constructive feedback.
>
> > Unlike the existing study [15, 42], the negative distribution is not considered.
>
> The method in [15] assumes the positive distribution shift as in our method. Thus, it does not consider the negative distribution change.
> The method in [42] assumes the covariate shift, i.e., $p _{{\rm te}} (x) \neq p _{{\rm tr}} (x)$ but $p _{{\rm te}}(y|x) = p _{{\rm tr}} (y|x)$. Although the covariate shift has been traditionally studied, the assumption of $p _{{\rm te}}(y|x) = p _{{\rm tr}} (y|x)$ is often restrictive in practice.
> One of the main contributions of our work is to highlight an important problem (both class imbalance and positive distribution shift occur) that has been overlooked despite its many applications, as described in Section 1.
>
> > The extension of the proposed method is discussed but not evaluated in the experiments.
>
> We have already evaluated the extension of our method (Eq. 16) in Table 6 of Section C.4.
> The extension of our method ($\alpha=0.999$) slightly tended to perform better than our original method ($\alpha=1.0$) by using additional labeled negative data.
>
> > Is the proposed loss function unbiased to its supervised counterpart?
>
> Yes. As long as the assumption of positive distribution shift (Eq. 7) is satisfied, the derived AUC in Eq. 12 is equivalent to the original AUC in Eq. 8.
>
> > It would be valuable if there were discussion or experimental results showing the effect of the number of unlabeled data from the test distribution.
>
> Thank you for the constructive comment. We additionally evaluate our method with a small number of
> unlabeled data from the test distribution.
> The results are described in Table 3 of the PDF file in the global response.
> As expected, the performance of our method tended to increase as the number of unlabeled data $N_{{\rm te}}$ increased. Nevertheless, our method tends to outperform puAUC, which does not use unlabeled test data, even when $N _{{\rm te}}=500$. Since many unlabeled data are often easy to collect, we believe that our method is useful in practice. We will include these results in the final version.
>
> > According to the literature, the non-negative risk estimator plays a crucial role in training deep neural networks. However, the proposed method does not mention the non-negativity of the risk estimator.
>
> Thank you for the insightful comment. In our preliminary experiment, we evaluated our method with the non-negative loss correction (Ours w/ nn), which prevents the empirical loss from being smaller than zero. However, the results with and without the loss correction (Ours w/ nn and Ours) were almost identical, as described in Table 4 of the PDF file. Thus, we proposed the current simpler loss function.
>
> We describe the non-negative correction used in this experiment.
> For clarity, we consider to minimize the AUC risk, $R _{\rho} (s) := \mathbb{E} _{{\bf x} ^{{\rm p}} \sim p ^{{\rm p}} _{{\rm te}} ({\bf x} )} \mathbb{E} _{{\bf x} ^{{\rm n}} \sim p ^{{\rm n}} _{{\rm te}} ({\bf x} )} [ f({\bf x} ^{{\rm n}}, {\bf x} ^{{\rm p}}) ] $, which is equivalent to maximize the AUC in Eq. 8 since  $ {\rm AUC} _{\rho} (s) = 1 - R _{\rho} (s) $.  The minimum value of this risk is zero. Then, the corresponding loss for Eq. 13 becomes ${\cal L} _{{\rm risk}}(s) :=  \mathbb{E} _{{\bf x} \sim p _{{\rm te}} ({\bf x}) } \mathbb{E} _{{\bar {\bf x} } \sim p _{{\rm tr} } ({\bf x}) } [ f({\bar {\bf x}} ,{\bf x}) ] - \pi _{{\rm tr} } \mathbb{E} _{{\bf x} \sim p _{{\rm te}} ({\bf x}) } \mathbb{E} _{{\bf x} ^{{\rm p}} \sim p ^{{\rm p}} _{{\rm tr}} ({\bf x}) } [ f({\bf x} ^{{\rm p}} ,{\bf x}) ] $.
> Since this loss is derived from the AUC risk, $\mathbb{E} _{{\bf x} \sim p _{{\rm te}} ({\bf x} )} \mathbb{E} _{{\bf x} ^{{\rm n}} \sim p ^{{\rm n}} _{{\rm te}} ({\bf x} )} [ f({\bf x} ^{{\rm n}}, {\bf x}) ]$, it should not take negative values. Thus, to prevent negative values of its empirical estimate ${\hat {\cal L} _{{\rm risk}}} (s) $, we used the loss with the absolute value function $|{\hat {\cal L} _{{\rm risk}}} (s)| $ for the optimization. This correction is successfully used in PU learning [15] or other weakly supervised learning [a].
>
> By the way, if class-prior $\pi _{{\rm te}}$ is known, we can derive a tighter lower bound on the AUC risk. Specifically, we can obtain $\mathbb{E} _{{\bf x} \sim p _{{\rm te}} ({\bf x} )} \mathbb{E} _{{\bf x} ^{{\rm n}} \sim p ^{{\rm n}} _{{\rm te}} ({\bf x} )} [ f({\bf x} ^{{\rm n}}, {\bf x}) ] \geq (1-\pi _{{\rm te}})/2$. Here, we used the definition of $p _{{\rm te}} ({\bf x} )$ and the fact that the AUC risk between the same densities is $1/2$ as described in Lines 163--164. By substituting Eq. 11 for this, we can obtain ${\cal L} _{{\rm risk}}(s) \geq (1-\pi _{{\rm te}})(1-\pi _{{\rm tr}})/2 =: b > 0$. As a result, we can use $| {\hat {\cal L} _{{\rm risk}}} (s) - b| + b$ for the optimization. Our method with this correction (Our w/ b) tended to enhance the performance of it without the correction (Ours) in Table 3. However, note that Ours has the strong advantage of not requiring the class-prior and performed better than existing methods. We will include this result in the final version.
>
> [a] Lu, Nan, et al. "Mitigating overfitting in supervised classification from two unlabeled datasets: A consistent risk correction approach." AISTATS2020.
>
> > Regarding the Extension in Section 4.4, it would be nice to cite the existing work.
>
> Thank you for the suggestion. Since the extension in Section 4.4 uses positive, negative, and unlabeled data in the training distribution, it is related to semi-supervised learning.
> A few studies [41,b] especially use the PU learning for semi-supervised learning.
> However, semi-supervised learning does not consider the distribution shift. The final version will include a more detailed discussion and citations.
>
> [b] Sakai et al. Semi-supervised classification based on classification from positive and unlabeled data. In ICML2017

---

> > ### Comment · Reviewer_4KuN · 2024-08-11
> >
> > Thank you for your answer.
> > Some of my questions and concerns are resolved.
> >
> > > The method in [42] assumes the covariate shift
> >
> > The paper [42] discussed the negative distribution shift and the risk to adapt the shift.
> >
> > > We have already evaluated the extension of our method (Eq. 16) in Table 6 of Section C.4.
> >
> > It would be better to put Table 6 in the main body.
> >
> > > Eq. 12 is equivalent to the original AUC in Eq. 8.
> >
> > It is nice to mention this for clarification.
> >
> > > the results with and without the loss correction (Ours w/ nn and Ours) were almost identical
> >
> > If there is an interpretation of why this happens, the contribution of this paper will become stronger.

---

> > > ### Author Response · Authors · 2024-08-12
> > >
> > > We appreciate your reply.
> > >
> > > > The paper [42] discussed the negative distribution shift and the risk to adapt the shift.
> > >
> > > Thank you for pointing this out. As you mentioned, the paper [42] discussed the negative distribution shift as well as the covariate shift.
> > > Specifically, this paper considers the case, $p _{{\rm te}} ^{{\rm p}} ({\bf x}) = p _{{\rm tr}} ^{{\rm p}} ({\bf x}) $ but $p _{{\rm te}} ^{{\rm n}} ({\bf x}) \neq p _{{\rm tr}} ^{{\rm n}} ({\bf x}) $.
> > > When positive and unlabeled data in the training distribution and unlabeled data in the test distribution are available, this paper showed that the original PU learning without re-weighting can deal with the negative distribution shift. This method requires class-prior $\pi _{{\rm te}}$.
> > >
> > > We illustrate that the same can be said for AUCs easily. That is, negative distribution shifts can be addressed by the existing AUC maximization studies [41,54,55]. Specifically, the AUC in the test distribution is
> > >
> > > ${\rm AUC}_{\sigma} (s) = \mathbb{E} _{{\bf x} ^{{\rm p}} \sim p ^{{\rm p}} _{{\rm te}} ({\bf x} )} \mathbb{E} _{{\bf x} ^{{\rm n}} \sim p ^{{\rm n}} _{{\rm te}} ({\bf x} )} \left[ f({\bf x} ^{{\rm p}}, {\bf x} ^{{\rm n}}) \right] = \frac{1}{1-\pi _{{\rm te}}} \mathbb{E} _{{\bf x} ^{{\rm p}} \sim p ^{{\rm p}} _{{\rm tr}} ({\bf x} )} \mathbb{E} _{{\bf x} \sim p _{{\rm te}} ({\bf x} )} \left[ f({\bf x} ^{{\rm p}}, {\bf x}) \right] + C$,
> > >
> > > where $C$ is a constant, and in the second equal sign, we used $p^{{\rm n}} _{{\rm te}} ({\bf x} ) = \frac{1}{1-\pi _{{\rm te}}} [ p _{{\rm te}}({\bf x}) - \pi _{{\rm te}} p _{{\rm te}} ^{{\rm p}}({\bf x}) ]$ and $p _{{\rm te}} ^{{\rm p}} ({\bf x}) = p _{{\rm tr}} ^{{\rm p}} ({\bf x})$, and the fact that the AUC between the same densities is a constant as described in Lines 163--164. The form of the derived AUC is equivalent to that in existing studies [41,54,55] and can be maximized with test unlabeled and positive training data. Unlike ordinary PU learning, test class-prior $\pi _{{\rm te}}$ is not required since it does not affect the optimization.
> > >
> > > We will include this discussion in the final version.
> > >
> > > > It would be better to put Table 6 in the main body.
> > >
> > > Thank you for the suggestion. We will put Table 6 in the main body of the final version.
> > >
> > > > It is nice to mention this for clarification.
> > >
> > > We agree on you. We will mention this in the final version for clarity.
> > >
> > > > If there is an interpretation of why this happens, the contribution of this paper will become stronger.
> > >
> > > The reason of the ineffectiveness of Ours w/ nn is that the non-negative constraint is insufficient/weak in our setting. Specifically, in Ours w/ nn, the non-negativity of the empirical loss, ${\hat {\cal L} _{{\rm risk}}} (s)$, is derived from the non-negativity of the corresponding expected loss, ${\cal L} _{{\rm risk}} (s) = (1-\pi _{{\rm tr}}) \mathbb{E} _{{\bf x} \sim p _{{\rm te}} ({\bf x} )} \mathbb{E} _{{\bf x} ^{{\rm n}} \sim p ^{{\rm n}} _{{\rm te}} ({\bf x} )} [ f({\bf x} ^{{\rm n}}, {\bf x}) ]$.
> > >
> > > Here, $f({\bf x} ^{{\rm n}}, {\bf x}) := \sigma (s({\bf x} ^{{\rm n}}) - s({\bf x})) \geq 0$, and when $s({\bf x}) \gg s({\bf x} ^{{\rm n}})$, $f({\bf x} ^{{\rm n}}, {\bf x})$ becomes zero.
> > > However, since $p _{{\rm te}} ^{{\rm n}} ({\bf x})$ is contained in $p _{{\rm te}} ({\bf x}) (= \pi _{{\rm te}} p _{{\rm te}} ^{{\rm p}} ({\bf x}) +  (1-\pi _{{\rm te}}) p _{{\rm te}} ^{{\rm n}} ({\bf x})$), the minimum value of the expected loss ${\cal L} _{{\rm risk}} (s)$ actually be greater than zero. Thus, the empirical loss could not be sufficiently constrained with the non-negativity, and thus, the performance did not improve.
> > >
> > > Additional information is required to know better/tighter constraints of the expected loss ${\cal L} _{{\rm risk}} (s)$.
> > > When class-prior in the test distribution $\pi _{{\rm te}}$ is known, the tighter constraint can be derived, ${\cal L} _{{\rm risk}}(s) \geq (1-\pi _{{\rm te}})(1-\pi _{{\rm tr}})/2 =: b > 0$, and using this (Ours w/ b) tends to enhance the performance of Ours in Table 3 in the PDF file. (Note that Ours has the advantage of not requiring $\pi _{{\rm te}}$ and performed better than the existing methods even without the loss correction).
> > >
> > > In the final version, we will include this discussion of why non-negativity is ineffective in our method.

---

> > > > ### Comment · Reviewer_4KuN · 2024-08-12
> > > >
> > > > Thak you for your answer.
> > > >
> > > > I still have questions about the discussion `The reason of the ineffectiveness of ...`.
> > > > According to [22], even though the expected risk is non-negative, the empirical risk becomes negative.
> > > > The PU learning method without risk correction performed poorly [22]. I was wondering why this does not happen with the proposed method.

---

> ### Author Response · Authors · 2024-08-13
>
> Thank you for your response.
>
> We investigated the learning process for the training loss, validation loss, and test AUC. Here, we used ${\hat {\cal L} _{{\rm risk}}} (s)$ for the loss. Note that the validation loss is the value of ${\hat {\cal L} _{{\rm risk}}} (s)$ calculated with validation data (PU data in the training distribution and U data in the test distribution).
>
> The training loss of our method became smaller than the lower bound $b$ as learning progressed and took negative on simple datasets (MNIST and Fashion MNIST). The validation loss (test AUC) tended to decrease (increase) initially but gradually increased (decreased) or stopped improving as learning progressed. These trends are consistent with ordinary PU learning, such as the study [22].
>
> However, the validation loss and test AUC were well correlated, and our method was able to select good models by using early-stopping with the validation loss. This is one of the reasons for the good performance of our method without the loss correction. Early-stopping was effective in preventing overfitting in the PU learning context.
> Note that all methods in our experiments used early-stopping.
>
> We will include this result and discussion in the final version.

---

> > ### Author Response · Authors · 2024-08-13
> >
> > We would like to kindly inform you that we have sent a response to your question. We may have posted it just as the OpenReview email notifications temporarily ceased, so we wanted to ensure you received it by notifying you again. We apologize for the inconvenience and appreciate your understanding.

---

> > > ### Comment · Reviewer_4KuN · 2024-08-13
> > >
> > > Thank you for your answers.
> > >
> > > The results in Fig. 2 in [22] show that even if we use early stopping, the test error of uPU at the best point sometimes has a gap from that of nnPU. If there are the results of the learning curve, like [22], the discussion in this thread will be more convincing.

---

> > > > ### Author Response · Authors · 2024-08-13
> > > >
> > > > Thank you for your response.
> > > >
> > > > > If there are the results of the learning curve, like [22], the discussion in this thread will be more convincing.
> > > >
> > > > We agree on this. However, according to the NeurIPS rebuttal instructional email, links to external sites are not allowed, so we cannot present figures like [22]. Thus, please let me continue with the textual explanation.
> > > >
> > > > > The results in Fig. 2 in [22] show that even if we use early stopping, the test error of uPU at the best point sometimes has a gap from that of nnPU.
> > > >
> > > > This is correct. The same thing was observed in our experiments: our method with the appropriate loss correction (Ours w/ b) tended to have a higher test AUC than it without the loss correction (Ours) at the best point.
> > > >
> > > > As previously mentioned in this thread, the test AUC of our method with and without the non-negative correction did not differ significantly. This is because the minimum value of our expected loss ${\cal L} _{{\rm risk}} (s)$ is actually greater than zero, which is theoretically derived in the previous response. Thus, the non-negative correction is insufficient for our loss.
> > > >
> > > > On the other hand, the non-negative correction was effective in ordinary PU learning [22] because the minimum value of the risk $R _{{\rm n}} ^{-} (g) := \mathbb{E} _{{\bf x} \sim p ^{\rm n} ({\bf x}) } [ \ell (g({\bf x}), -1)]$ used in [22] is zero, where $\ell$ is a point-wise loss and $g$ is a decision function.
> > > > This creates a gap in the effectiveness of the non-negative correction in our paper and in [22].
> > > >
> > > > We would like to include the figures and this discussion in the final version.

---

> > > > > ### Comment · Reviewer_4KuN · 2024-08-13
> > > > >
> > > > > Thank you for your answers. The textual explanation is fine.
> > > > >
> > > > > I sincerely thank the authors for their humble and respectful responses during discussions.
> > > > > If the answers, results, and figures are really included in the final version, my final score will be five.

---

> > > > > > ### Author Response · Authors · 2024-08-13
> > > > > >
> > > > > > Thank you very much for your response. We are pleased to learn that you have increased the rating to 5. We will ensure that the content, results, and figures from our discussions are incorporated into the final version. Your feedback has undoubtedly contributed to the improvement of our work.
> > > > > > We sincerely appreciate the time and effort you have devoted to reviewing and discussing our paper.

---

### Official Review · Reviewer_QCqD · 2024-07-13

**Soundness:** 3
**Presentation:** 3
**Contribution:** 2
**Rating:** 4
**Confidence:** 4

**Summary:**

This paper addresses the challenge of maximizing the Area Under the Receiver Operating Characteristic Curve (AUC) in imbalanced binary classification problems where there is a positive distribution shift--this shift is where negative data remains constant, but positive data varies. A new method is proposed that utilizes labeled positive and unlabeled data from the training distribution, along with unlabeled data from the test distribution, to maximize the AUC effectively in the presence of such shifts.

**Strengths:**

This paper introduces a new loss function designed for AUC maximization under positive distribution shifts. Previous research has focused separately on AUC maximization and positive distribution shifts, but this study found the intersection of these two areas. The authors have successfully identified and explored this new research niche. The proposed loss function, derived from mathematical foundations, can be readily integrated into neural network training, offering a practical application for enhancing model performance. This paper is well-structured and clearly written, making it easy to follow.

**Weaknesses:**

Despite its strengths, this research primarily offers a simple proposal of a loss function, suggesting its contributions to the field might be limited. An expansion to include various metrics, such as F-1 and G-mean of TPR and TNR, which are also relevant for imbalanced data classification, could enrich this paper. Additionally, the experimental validation is somewhat restricted, utilizing only four datasets, all of which are image datasets. A more comprehensive evaluation using a broader range of datasets is necessary to fully assess the proposed loss function's effectiveness. Therefore, the reviewer believes that the contribution of this research may not be substantial enough for acceptance at a top-tier conference.

**Questions:**

Q1: Please specify the scenarios where both class imbalance and positive distribution shift occur. Providing detailed examples will help readers grasp the practical significance of this research problem.

Q2: Why did the authors choose to conduct their experiments exclusively with image datasets? Are there any other real-world problems?

Q3: AUC maximization can be implemented not just as a loss function for neural networks, but across various machine learning methods. Why did you choose to focus on proposing a loss function?

Q4: The reviewer is not convinced that Lines 4-5 in Algorithm 1 sufficiently demonstrate the training process. There needs to be a more detailed and mathematical explanation of how model parameters are updated using the proposed loss function.

**Limitations:**

L1: An expansion to include various metrics, such as F-1 and G-mean of TPR and TNR, which are also relevant for imbalanced data classification, could enrich this paper.

L2: For extremely imbalanced cases, training difficulties are likely to arise. It is necessary to address how the proposed loss function can be effectively minimized in these scenarios.

---

> ### Author Rebuttal · Authors · 2024-08-06
>
> Thank you for your insightful comments and constructive feedback.
>
> > An expansion to include various metrics, such as F-1 and G-mean of TPR and TNR, which are also relevant for imbalanced data classification, could enrich this paper.
>
> Thank you for the suggestion. We agree that extensions to maximize other metrics such as F-1 could enrich our paper. However, AUC is the most representative metric for imbalanced data, and many papers (including top conference and journal papers such as ICLR, AAAI, ICCV, ICDM, NeurIPS, and TPAMI) focus solely on AUC maximization [12, 29, 41, 54, 55, 57, 58, 60, 62]. We, therefore, believe that the current proposal is still worthy of publication. Extending our problem setting to maximize other metrics is an interesting future work.
>
> As a supplement, we evaluated F1 scores of each method.
> The results are described in Table 1 of the PDF file in the global response.
> Our method also outperformed the others even when it maximizes the AUC.
> This result may imply that AUC maximization can also help improve other evaluation metrics for imbalanced data.
> We will include this result and the above discussion (future work) in the final version.
>
> > the experimental validation is somewhat restricted, utilizing only four datasets, all of which are image datasets. A more comprehensive evaluation using a broader range of datasets is necessary to fully assess the proposed loss function's effectiveness.
>
> Thank you for the constructive comment.
> We additionally evaluate our method with two tabular datasets (Hospital Readmission and Hypertension [a]).
> In Hospital Readmission, the task is to predict the 30-day readmission of diabetic hospital patients.
> In Hypertension, the task is a hypertension diagnosis for high-risk age.
> Both datasets are widely used for distribution shift adaptation studies [a].
> We used positive-shifted data to construct the positive distribution shift.
> The experimental settings, such as the number of data, are the same as the submitted paper.
> The results are described in Table 2 of the PDF file.
> Our method outperformed the others. These results show that our method works well in tabular datasets. We will include these results in the final version.
>
> [a] Gardner et al. "Benchmarking distribution shift in tabular data with tableshift." NeurIPS2023.
>
> > Q1: Please specify the scenarios where both class imbalance and positive distribution shift occur. Providing detailed examples will help readers grasp the practical significance of this research problem.
>
> We have described the motivating examples (cyber security, medical diagnosis, and visual inspection) where both class imbalance and positive distribution shifts occur in the first and second paragraphs of Section 1. For example, in cyber security, malicious data (positive data) are much smaller than benign data (negative data). In addition, malicious adversaries often change their attacks (malicious data) to bypass detection systems, while benign data do not change much [9, 15, 63]. Thus, our problem setting is suitable for this application.
>
> > Q3: AUC maximization can be implemented not just as a loss function for neural networks, but across various machine learning methods. Why did you choose to focus on proposing a loss function?
>
> We focus on the loss function because it has a high degree of generality: it can be combined with any (differentiable) models such as linear models, kernel models, and neural networks, as described in Lines 66--67. This characteristic is beneficial in practice because the available computing resources vary depending on the application site.
> For example, when resources such as GPUs are scarce, which is often the case, our loss function can be used with lightweight models such as liner models. When ample resources are available, huge models such as large neural networks can also be used.
>
> > Q4: The reviewer is not convinced that Lines 4-5 in Algorithm 1 sufficiently demonstrate the training process. There needs to be a more detailed and mathematical explanation of how model parameters are updated using the proposed loss function.
>
> Sorry for the lack of explanation.
> We describe a more detailed and mathematical explanation of Lines 4 and 5 in Algorithm 1 below:
>
> - Let $\\{ {\bar x} _{{\rm tr},m} ^{\rm p} \\} _{m=1}^{P}$ be sampled positive data from $X _{{\rm tr}} ^{\rm p}$ and $\\{ {\bar x} _{{\rm tr},m} \\} _{m=1} ^{M _{{\rm tr}}} \cup \\{ {\bar x} _{{\rm te},m} \\} _{m=1} ^{M _{{\rm te}}}$ be sample unlabeled data from $X _{{\rm tr}} \cup X _{{\rm te}}$. Then, in Line 4, we calculate the loss in Eq. 14 on the sampled data. That is,
> ${\hat {\mathcal L}} _{{\rm sampled}} (\theta) := - \frac{1}{M _{{\rm te}} M _{{\rm tr}}} \sum _{n,m=1} ^{M _{{\rm te}}, M _{{\rm tr}}} f({\bar {\bf x}} _{{\rm te}, n}, {\bar {\bf x}} _{{\rm tr}, m}) + \frac{\pi _{{\rm tr}}}{M _{{\rm te}} P} \sum _{n,m=1} ^{M _{{\rm te}}, P } f({\bar {\bf x}} _{{\rm te}, n}, {\bar {\bf x}} _{{\rm tr}, m} ^{{\rm p}})$,
> where $\theta$ is parameters of score function $s$.
>
> - In Line 5, we update $\theta$ by using a stochastic gradient descent. That is,
> $\theta \leftarrow \theta - \mu \frac{\partial {\hat {\mathcal L}} _{{\rm sampled}}}{\partial \theta} (\theta)$,
> where $\mu \in \mathbb{R} _{>0}$ is a learning rate.
>
> We will add this explanation in the final version.
>
> > L2: For extremely imbalanced cases, training difficulties are likely to arise. It is necessary to address how the proposed loss function can be effectively minimized in these scenarios.
>
> As you mentioned, highly imbalanced data will be difficult to learn for all methods, including ours. This is due to the difficulty in extracting positive data information from unlabeled data. However, in Table 4 of Section C.2, we confirmed that our method outperformed the others when small class-priors on the training distribution (e.g., $\pi _{{\rm tr}}=0.01$) are used.

---

### Author Rebuttal · Authors · 2024-08-06

Dear all reviewers,

Thank you very much for the detailed and constructive feedback on our paper. We would like to revise the paper based on the comments. A pdf file with additional experiments is attached to this global response.

Best regards,

Authors

---

### Decision · Program_Chairs · 2024-09-25

**Decision:**

Accept (poster)

**Comment:**

The addressed task is very relevant in practice. One of the reviewers was a bit unhappy that it only applies for AUC. It would indeed be useful to know about whether similar approaches could be developed for other objectives, but it is perfectly fine that the paper addresses only AUC. The appendix includes information about F1 as well, but this is just informative, not stating that the method should be applied to improve F1 when there is positive distribution shift.

The proposed method is quite simple and uses a trick that is somewhat similar to many other methods in transfer learning, but this is rather a strength not a weakness.

One of the reviewers (4KuN) had first lower grades for presentation, soundness, and contribution, but raised the overall rating from 4 to 5 after a lengthy discussion with the authors, stating eventually that "The textual explanation is fine".

One of the most positive reviewers (CKbq) claimed that "The authors do not fully compare their method with the latest ones" and gave 3 references. However, these papers did not address the shift on the positives as the current paper does. Regardless, in the rebuttal phase, the authors included one of those methods into the comparison. As expected, that method was outperformed by the method proposed in the current paper.

The reviewers had no major criticisms about the quality of presentation.

Therefore, I recommend the paper for acceptance.